# Advantage-Guided Distillation for Preference Alignment in Small Language Models

**Shiping Gao**[1], **Fanqi Wan**[1], **Jiajian Guo**[1], **Xiaojun Quan**[1*], **Qifan Wang**[2]
[1]School of Computer Science and Engineering, Sun Yat-sen University, China
[2]Meta AI
{gaoshp,wanfq,guojj59}@mail2.sysu.edu.cn
quanxj3@mail.sysu.edu.cn
wqfcr@fb.com

## Abstract

Alignment techniques enable large language models (LLMs) to generate outputs that align with human preferences and play a crucial role in their effectiveness. However, their impact often diminishes when applied to small language models (SLMs), likely due to the limited capacity of these models. Instead of directly applying existing alignment techniques to SLMs, we propose to utilize a well-aligned teacher LLM to guide the alignment process for these models, thereby facilitating the transfer of the teacher's knowledge of human preferences to the student model. To achieve this, we first explore a straightforward approach, Dual-Constrained Knowledge Distillation (DCKD), that employs knowledge distillation with two KL-divergence constraints from the aligned teacher to the unaligned student. To further enhance the student's ability to distinguish between preferred and dispreferred responses, we then propose Advantage-Guided Distillation for Preference Alignment (ADPA), which leverages an advantage function from the aligned teacher to deliver more nuanced, distribution-level reward signals for the student's alignment. Our experimental results show that these two approaches appreciably improve the alignment of SLMs and narrow the performance gap with larger counterparts. Among them, ADPA demonstrates superior performance and achieves even greater effectiveness when integrated with DCKD. Our code is available at https://github.com/SLIT-AI/ADPA.

## 1 Introduction

Large language models (LLMs) can be effectively aligned with human preferences to generate helpful, truthful, and harmless responses using techniques like reinforcement learning from human feedback (RLHF) (Kaplan et al., 2020; Ouyang et al., 2022; Askell et al., 2021). However, deploying such large models in resource-constrained environments can be challenging due to their heavy computational and memory demands. While small language models (SLMs) are more suitable for these scenarios, they often struggle to achieve the same level of alignment as larger LLMs. These small models may experience an "alignment tax", where their overall performance across various tasks declines after RLHF training (Bai et al., 2022). This decline is likely due to their limited capacity to capture the complexities of diverse tasks and nuanced human feedback, which can result in overfitting and poor generalization (Kirk et al., 2024; Zhao et al., 2023a). Moreover, traditional RLHF methods depend on sequence-level rewards, which are sparse and coarse-grained (Sun, 2023; Chan et al., 2024), posing greater optimization challenges for SLMs.

To enhance the alignment of SLMs with human preferences and achieve an "alignment bonus", a promising strategy is to leverage preference-aligned larger models to guide smaller models through knowledge distillation (KD) (Hinton, 2015). KD enables the student model to learn from the teacher's predictions and internal representations, which provide nuanced learning signals (Gu et al., 2024), effectively transferring knowledge from teacher to student. However, existing KD methods primarily focus on the pre-training and instruction-tuning stages (Song et al., 2020; Khanuja et al.,

---
*Corresponding author.

2021) and often overlook the critical phase of preference alignment. This oversight prevents student models from capturing the teacher's alignment knowledge with human preferences. Moreover, most KD techniques emphasize positive signals from the teacher's outputs on ground-truth responses while neglecting negative signals from suboptimal outputs, which limits the overall alignment effect. Fortunately, these issues have recently garnered attention from the community. For instance, DPKD (Li et al., 2024) and PLaD (Zhang et al., 2024) treat the teacher's outputs as preferred responses and the student's outputs as dispreferred and carry out preference learning to train the student model.

In this work, we first explore a straightforward approach, Dual-Constrained Knowledge Distillation (DCKD), for aligning SLMs with human preferences. This method enables direct knowledge distillation from an aligned teacher model to an unaligned student model using preference training data. To incorporate both positive and negative signals, we introduce an additional KL-divergence constraint term for dispreferred responses into the traditional knowledge distillation objective. This allows the student model to capture the teacher's predictive behaviors for both preferred and dispreferred responses. However, while this approach facilitates the direct transfer of preference knowledge from teacher to student models, its effectiveness may be limited by the lack of a contrastive mechanism during training to better distinguish between preferred and dispreferred responses.

To overcome this limitation, we further propose Advantage-Guided Distillation for Preference Alignment (ADPA). ADPA introduces stronger contrastive signals by incorporating a fine-grained preference alignment mechanism into the distillation process, allowing the teacher model to better guide the student model during training. Specifically, ADPA utilizes an advantage function derived from a teacher model trained with Direct Preference Optimization (DPO) (Rafailov et al., 2024b) and a pre-DPO reference teacher model. The advantage function delivers distribution-level reward signals and allows the student model to optimize its policy based on fine-grained preference signals, which also tackles the issue of sparse reward signals present in traditional RLHF. Moreover, the effect of preference-guided distillation in ADPA can be amplified by leveraging a DCKD-initialized student model, leading to an enhanced variant called ADPA+. As illustrated in Figure 1, ADPA+ enables smaller models to capture human preferences more effectively than directly applying DPO, narrowing the performance gap with larger counterparts.

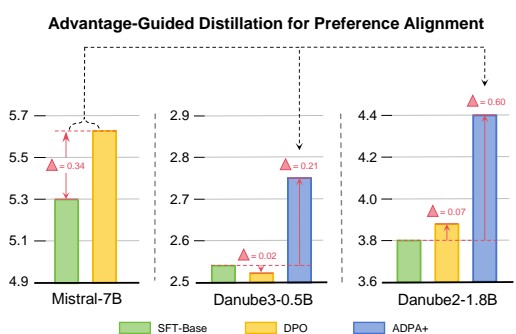

Figure 1: Preliminary results showing the "alignment tax" on smaller models and the impact of our ADPA+ method on MT-Bench (Zheng et al., 2023) rating. Under DPO training, the larger model (Mistral-7B) improves notably (+0.34), while smaller models show limited gains (+0.07 for Danube2-1.8B) or even a drop (-0.02 for Danube3-500M). In contrast, ADPA+ enables small models to achieve greater gains (+0.21 for Danube3-500M and +0.60 for Danube2-1.8B).

The major contributions of this work can be summarized as follows:

- We investigate the alignment challenge for small language models (SLMs) through knowledge distillation (KD) from a preference-aligned teacher to a smaller student model. We introduce Dual-Constrained Knowledge Distillation (DCKD) as a baseline approach.

- We propose Advantage-Guided Distillation for Preference Alignment (ADPA), which utilizes an advantage function from a preference-aligned teacher and a reference teacher to provide distribution-level reward signals for optimizing the student model.

- We conduct extensive experiments to demonstrate the effectiveness of our proposed approaches and provide valuable insights into aligning SLMs with human preferences. Specifically, leveraging preference-aligned larger models to guide SLM alignment training shows great promise in addressing their limited capacity and enhancing alignment effectiveness.

## 2  RELATED WORK

**Knowledge Distillation**   Knowledge distillation (KD) (Hinton, 2015) is a popular model compression technique where a small student model learns to mimic the output distributions, hidden

layer outputs (Chen et al., 2017), inter-layer relationships (Yim et al., 2017), sample relationships (Reddi et al., 2021), or attributions (Wu et al., 2023) of one or more larger teacher models (Liu et al., 2020). In the context of LLMs, KD typically focuses on minimizing the Kullback-Leibler Divergence (KLD) between the output distributions of the student and teacher models at each time step. Recent research has proposed various optimizations to improve the efficiency and effectiveness of this process. For instance, MiniLLM (Gu et al., 2024) employs sequence-level reverse KLD to encourage the student model to focus on the most significant modes of the teacher's output distributions. DistiLLM (Ko et al., 2024), on the other hand, introduces a novel skew KLD objective, which interpolates between the teacher's and student's distributions to ensure stable gradients and reduce optimization errors. Likewise, $f$-distill (Wen et al., 2023) minimizes a symmetric $f$-divergence to mitigate challenges such as mode collapse, while Adaptive Kullback-Leiber (AKL) (Wu et al., 2024) balances forward and reverse KLD to ensure the student model effectively learns across different parts of the distribution. Other approaches, including Vicuna (Chiang et al., 2023) and MCC-KD (Chen et al., 2023), take advantage of outputs generated by the teacher model to train the student, thereby enhancing its ability to follow instructions or perform more complex reasoning tasks.

**Preference Alignment** Preference alignment aims to align the outputs of LLMs with human preferences and values. This objective is traditionally achieved by RLHF (Ouyang et al., 2022), which relies on a reward model (RM) trained on preference data to guide the optimization of the policy model through policy gradient optimization methods like Proximal Policy Optimization (PPO) (Schulman et al., 2017). Recent research has increasingly focused on using contrastive learning methods to eliminate the need of RM and complex online reinforcement learning (RL). Notable approaches in this area include Direct Preference Optimization (DPO) (Rafailov et al., 2024b) and SLiC-HF (Zhao et al., 2023b). Other studies explore fine-grained rewards to provide more detailed guidance to the policy model. For example, Yang et al. (2023b) defined sequence-level rewards as aggregations of token-wise rewards learned through sequence-level RM training on preference datasets. In addition, Token-Level Continuous Reward (TLCR) (Yoon et al., 2024) employs GPT-4 as a reviser to analyze preference pairs and modify dispreferred responses to generate token-level preference labels that are then used to train a discriminator for assigning token-level rewards.

Given the high cost of obtaining quality preference labels for training reward models, recent research has focused on leveraging larger and more powerful LLMs to provide feedback on the preferences of candidate responses. For example, RLAIF (Lee et al., 2023) uses an off-the-shelf LLM to offer feedback on candidate responses, which are then used to train a reward model for RL. Zephyr (Tunstall et al., 2023) and Starling (Zhu et al., 2024) gather responses from multiple LLMs and rank them using GPT-4 to obtain preference data. While the former uses this data to train a policy with DPO, the latter employs it to train a reward model for RL. Other approaches, such as DPKD (Li et al., 2024) and PLaD (Zhang et al., 2024), treat the teacher's outputs as preferred responses and the student's outputs as dispreferred and conduct preference learning. RLCD (Yang et al., 2023a) designs positive and negative prompts to elicit corresponding responses and uses them to train a reward model for RL. Reward model distillation (Fisch et al., 2024) aligns the distribution predicted by the policy with that of a trained reward model to enhance preference optimization robustness.

## 3 METHODOLOGY

In this section, we present our proposed methods for enhancing preference alignment in SLMs: Dual-Constrained Knowledge Distillation (DCKD) and Advantage-Guided Distillation for Preference Alignment (ADPA). We start with an overview of the preliminaries of knowledge distillation and preference alignment in LLMs. We then detail the DCKD and ADPA methods, highlighting how ADPA addresses the limitations of DCKD to achieve better alignment with human preferences.

### 3.1 PRELIMINARIES

**Knowledge Distillation** Given a dataset of prompt-response pairs $(x, y)$, a teacher LLM $\pi_t$, and a smaller student model $\pi_\theta$, the goal of knowledge distillation (KD) is to train the student model to mimic the teacher's predictions as closely as possible. Typically, the objective function comprises two loss terms. The first is the supervised fine-tuning (SFT) loss: $\mathcal{L}_{\text{SFT}} = -\frac{1}{|y|} \sum_{t=1}^{|y|} \log \pi_\theta(y_t \mid x, y_{<t})$, which computes the negative log-likelihood (NLL) of the student model predicting the next token $y_t$ in the response, conditioned on the prompt $x$ and previously-generated response tokens $y_{<t}$. The second term is the Kullback-Leibler Divergence (KLD) between the output distributions

of the teacher and the student models. These two terms are combined using a weighted sum:

$$\mathcal{L}_{\text{KD}} = -\frac{1}{|y|} \sum_{t=1}^{|y|} \left[ \log \pi_\theta(y_t \mid x, y_{<t}) + \alpha D_{\text{KL}} \left( \pi_t(\cdot \mid x, y_{<t}) \mid\mid \pi_\theta(\cdot \mid x, y_{<t}) \right) \right]. \quad (1)$$

**Preference Alignment for LLMs**   Preference alignment methods such as reinforcement learning from human feedback (RLHF) (Ouyang et al., 2022) optimize LLMs to produce outputs that align with human preferences. Given a preference dataset $\mathcal{D}$ containing triples of prompt $x$, preferred response $y_w$, and dispreferred response $y_l$, a sequence-level reward model (RM) can be trained as:

$$\mathcal{L}_{\text{RM}} = -\mathbb{E}_{(x,y_w,y_l)\sim\mathcal{D}} \left[ \log \sigma \left( \text{RM}_\phi(x, y_w) - \text{RM}_\phi(x, y_l) \right) \right], \quad (2)$$

where $\sigma$ is the sigmoid function. After training the RM, classical RLHF methods optimize the SFT-trained LLMs using policy gradient techniques like Proximal Policy Optimization (PPO) (Schulman et al., 2017). Formally, the objective is to maximize the sequence-level reward assigned by the RM while penalizing deviations from a reference policy using a KLD term, weighted by a coefficient $\beta$:

$$\max_\theta \mathbb{E}_{y\sim\pi_\theta(\cdot|x)} \left[ \text{RM}(x, y) - \beta \log \frac{\pi_\theta(y \mid x)}{\pi_{\text{ref}}(y \mid x)} \right], \quad (3)$$

where $\pi_{\text{ref}}$ denotes the reference policy. Offline RLHF methods like Direct Preference Optimization (DPO) (Rafailov et al., 2024b) directly optimize the policy model within the Bradley-Terry modeling framework (Bradley & Terry, 1952). Unlike traditional RLHF approaches, DPO eliminates the need for an explicit external reward model or online reinforcement learning, instead leveraging preference data directly to align the model's outputs with human feedback:

$$\mathcal{L}_{\text{DPO}}(\pi_\theta, \pi_{\text{ref}}) = -\mathbb{E}_{(x,y_w,y_l)\sim\mathcal{D}} \left[ \log \sigma \left( \beta \log \frac{\pi_\theta(y_w \mid x)}{\pi_{\text{ref}}(y_w \mid x)} - \beta \log \frac{\pi_\theta(y_l \mid x)}{\pi_{\text{ref}}(y_l \mid x)} \right) \right]. \quad (4)$$

### 3.2   DUAL-CONSTRAINED KNOWLEDGE DISTILLATION

A straightforward approach to transferring preference knowledge from large models to smaller ones is to perform knowledge distillation using preference data. Our Dual-Constrained Knowledge Distillation (DCKD) method operates by first fine-tuning the teacher model on preference data using DPO, yielding a teacher policy $\pi_{\text{dpo}}$ that captures human preferences. The distillation process then minimizes the divergence between the output distributions of the teacher and student models for both preferred and dispreferred responses. Formally, for each pair of responses $(y_w, y_l)$, where $y_w$ is preferred and $y_l$ is dispreferred, we define two KL-divergence constraints:

$$\mathcal{L}_{\text{KLD-}w}(\pi_{\text{dpo}}, \pi_\theta) = \mathbb{E}_{(x,y_w)\sim\mathcal{D}} \left[ \sum_{t=1}^{|y_w|} D_{\text{KL}} \left( \pi_{\text{dpo}}(\cdot \mid x, y_{w,<t}) \mid\mid \pi_\theta(\cdot \mid x, y_{w,<t}) \right) \right], \quad (5)$$

$$\mathcal{L}_{\text{KLD-}l}(\pi_{\text{dpo}}, \pi_\theta) = \mathbb{E}_{(x,y_l)\sim\mathcal{D}} \left[ \sum_{t=1}^{|y_l|} D_{\text{KL}} \left( \pi_{\text{dpo}}(\cdot \mid x, y_{l,<t}) \mid\mid \pi_\theta(\cdot \mid x, y_{l,<t}) \right) \right]. \quad (6)$$

Including the SFT term on preferred responses, the overall objective of DCKD is:

$$\mathcal{L}_{\text{DCKD}} = \mathcal{L}_{\text{SFT}} + \alpha \left( \mathcal{L}_{\text{KLD-}w} + \mathcal{L}_{\text{KLD-}l} \right). \quad (7)$$

DCKD differs from traditional knowledge distillation in two key aspects. First, it distills from a preference-aligned teacher model fine-tuned with DPO, which gives richer preference information. Second, it minimizes the KL-divergence for both preferred and dispreferred responses, enabling the student model to better understand the nuances of human preferences.

### 3.3   ADVANTAGE-GUIDED DISTILLATION FOR PREFERENCE ALIGNMENT

While DCKD enables direct transfer of preference knowledge from the teacher to the student, it focuses on aligning the student's output distributions with those of the teacher. As a result, it may not provide enough guidance for the student to distinguish between positive and negative responses.

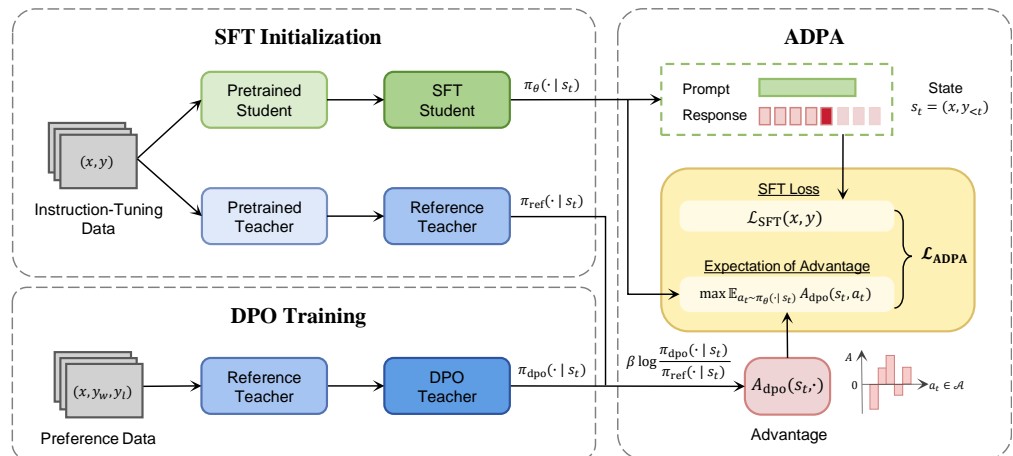

Figure 2: Overview of ADPA. Training involves two teacher models: a DPO teacher $\pi_{\mathrm{dpo}}$ fine-tuned on preference data and a reference teacher $\pi_{\mathrm{ref}}$ fine-tuned on instruction-tuning data. The student model is trained via instruction-tuning and then advantage-guided distillation with on-policy data.

To address this limitation, we propose Advantage-Guided Distillation for Preference Alignment (ADPA), as illustrated in Figure 2. The core idea of ADPA is to leverage the difference between the DPO-trained teacher model (DPO teacher) $\pi_{\mathrm{dpo}}$ and the pre-DPO reference teacher model (reference teacher) $\pi_{\mathrm{ref}}$ to derive an advantage function. This advantage function quantifies the relative preference of each action, highlighting how much more (or less) the DPO teacher favors an action compared to the reference teacher. By focusing on this difference, we provide the student model with explicit and fine-grained signals about which actions are more aligned with human preferences.

**Deriving the Advantage Function** Given the reference teacher $\pi_{\mathrm{ref}}$ and the DPO teacher model $\pi_{\mathrm{dpo}}$ (initialized from $\pi_{\mathrm{ref}}$), let $s_t = (x, \hat{y}_{<t})$ represent the state at time step $t$, comprising the prompt $x$ and the response tokens $\hat{y}_{<t}$ generated so far. Furthermore, let $\mathcal{A}$ denote the action set (or vocabulary), and let $a_t \in \mathcal{A}$ represent the action (or token) at time step $t$. The advantage function is derived from $\pi_{\mathrm{dpo}}$ and $\pi_{\mathrm{ref}}$ as follows (Rafailov et al., 2024a):

$$A_{\mathrm{dpo}}(s_t, a_t) = \beta \log \frac{\pi_{\mathrm{dpo}}(a_t \mid s_t)}{\pi_{\mathrm{ref}}(a_t \mid s_t)}. \tag{8}$$

This advantage function shows how DPO training adjusts the teacher's action probabilities relative to the reference model, reflecting preference changes for action $a_t$ at state $s_t$ (proof in Appendix A).

**ADPA Training Objective** In our ADPA, the advantage function $A_{\mathrm{dpo}}$ is incorporated into the training objective to offer fine-grained guidance to the student policy. The goal is to maximize the expected advantage of the student policy as follows:

$$\max_{\theta} \mathbb{E}_{a_t \sim \pi_\theta(\cdot \mid s_t)} A_{\mathrm{dpo}}(s_t, a_t) = \max_{\theta} \mathbb{E}_{a_t \sim \pi_\theta(\cdot \mid s_t)} \beta \log \frac{\pi_{\mathrm{dpo}}(a_t \mid s_t)}{\pi_{\mathrm{ref}}(a_t \mid s_t)}, \tag{9}$$

where $s_t = (x, \hat{y}_{<t})$ and $\hat{y}$ is the response generated by the student model for prompt $x$.

Let $\hat{\mathcal{D}}$ denote the dataset containing prompts, ground truth responses, and student's generated responses, the overall ADPA loss function is defined as:

$$\mathcal{L}_{\mathrm{ADPA}} = \mathbb{E}_{(x,y,\hat{y}) \sim \hat{\mathcal{D}}} \left[ \mathcal{L}_{\mathrm{SFT}}(x, y) - \gamma \sum_{t=1}^{|\hat{y}|} \sum_{a_t \in \mathcal{A}} \pi_\theta(a_t \mid s_t) \log \frac{\pi_{\mathrm{dpo}}(a_t \mid s_t)}{\pi_{\mathrm{ref}}(a_t \mid s_t)} \right]. \tag{10}$$

Here, $y$ is the ground truth response for prompt $x$, $\mathcal{L}_{\mathrm{SFT}}(x, y)$ represents the supervised fine-tuning loss to preserve basic capabilities and avoid over-optimization (Liu et al., 2024b), and the hyperparameter $\gamma$ balances the supervised fine-tuning term and the advantage-guided distillation term.

The advantage function in ADPA provides distribution-level reward signals to the student policy, allowing it to be refined by focusing on the nuanced differences between positive and negative actions. It also relieves the issue of sparse reward signals typical in traditional RLHF.

**Gradient Analysis of the ADPA Loss Function**  To understand how ADPA guides the student policy, we analyze the gradient of the ADPA loss function with respect to the model parameters $\theta$. Ignoring the SFT loss for simplicity, the gradient of the advantage-guided distillation term is:

$$\nabla_\theta \mathcal{L}_{\text{ADPA w/o. SFT}} = -\gamma \sum_{t=1}^{|\hat{y}|} \sum_{a_t \in \mathcal{A}} \nabla_\theta \pi_\theta(a_t \mid s_t) \cdot A_{\text{dpo}}(s_t, a_t). \tag{11}$$

Since $\nabla_\theta \log \pi_\theta(a_t \mid s_t) = \frac{1}{\pi_\theta(a_t|s_t)} \nabla_\theta \pi_\theta(a_t \mid s_t)$, we derive the identity $\nabla_\theta \pi_\theta(a_t \mid s_t) = \pi_\theta(a_t \mid s_t) \nabla_\theta \log \pi_\theta(a_t \mid s_t)$. Utilizing this, we can rewrite the gradient in Eq. (11) as:

$$\nabla_\theta \mathcal{L}_{\text{ADPA w/o. SFT}} = -\gamma \sum_{t=1}^{|\hat{y}|} \mathbb{E}_{a_t \sim \pi_\theta(\cdot|s_t)} \left[ \nabla_\theta \log \pi_\theta(a_t \mid s_t) \cdot A_{\text{dpo}}(s_t, a_t) \right]. \tag{12}$$

This gradient closely resembles the policy gradient in reinforcement learning, where updates are weighted by the advantage function. First, the student model is guided to increase the probabilities of actions with positive advantage (preferred actions) and decrease the probabilities of those with negative advantage (dispreferred actions). Second, by weighting updates based on the magnitude of the advantage, the student receives fine-grained guidance that emphasizes relative preferences, captured by the difference between $\log \pi_{\text{dpo}}$ and $\log \pi_{\text{ref}}$. This ensures that the student model not only learns effectively from the teacher but also focuses on actions that better align with human preferences. Moreover, ADPA eliminates the need for online RL and enables more stable and efficient training of the student policy. Further analysis is provided in Section 4.4.

# 4 EXPERIMENT

## 4.1 SETUP

**Backbones and Datasets**  In our experiments, we evaluate the preference alignment performance of the proposed methods using four SLMs: H2O-Danube3-500M (Pfeiffer et al., 2024), H2O-Danube2-1.8B (Singer et al., 2024), LLaMA-2-7B, and LLaMA-3.2-1B. For the teacher models, we selected Mistral-7B-V0.1 (Jiang et al., 2023) for the first two students, LLaMA-2-13B (Touvron et al., 2023) for the third, and LLaMA-3.1-8B for the fourth. To ensure a solid starting point, we perform supervised fine-tuning (SFT) on both student and teacher models using the Deita-10K-V0 (Liu et al., 2024a) dataset, which comprises 10k high-quality instruction-response pairs. For preference alignment, we investigate two datasets: DPO-MIX-7K[1], a curated collection of high-quality pairwise comparison data, and HelpSteer2 (Wang et al., 2024), which is developed to align models for enhanced helpfulness. When using HelpSteer2, we differentiate between positive and negative samples based on the *helpfulness* metric, excluding those where the scores for both aspects are identical.

**Validation**  We employ FsfairX-LLaMA3-RM-V0.1[2] (hereafter referred to as FsfairX) to evaluate and select optimal checkpoints during training. FsfairX is a high-performing reward model on RewardBench (Lambert et al., 2024) and calculates average response scores for prompts in the validation set of each preference dataset (DPO-MIX-7K/HelpSteer2).

**Evaluation**  We evaluate model performance using three benchmarks: MT-Bench (Zheng et al., 2023), AlpacaEval (Li et al., 2023), and the Open LLM Leaderboard (Beeching et al., 2023). For MT-Bench, we use GPT-4-0125-Preview as the evaluator, following recent recommendations[3] to correct inaccuracies in GPT-4's original reference answers. In AlpacaEval, to avoid disadvantaging smaller models, we compare against ADPA-trained student models as references and compute *win rate* using GPT-4-1106-Preview as the evaluator. For the Open LLM Leaderboard, we adhere to the recommended Language Model Evaluation Harness (Gao et al., 2021) protocols.

**Baselines**  We compare DCKD, ADPA, and ADPA+ against three preference alignment methods—DPO (Rafailov et al., 2024b), SimPO (Meng et al., 2024), and WPO (Zhou et al., 2024)—as well as several state-of-the-art baselines, including vanilla knowledge distillation (VanillaKD) (Hinton, 2015), SeqKD (Kim & Rush, 2016), ATKD (Zhong et al., 2024b), PLaD (Zhang et al., 2024), DDPO (Fisch et al., 2024), and DPKD (Li et al., 2024). As introduced earlier, ADPA+ initializes

---

[1] https://huggingface.co/datasets/argilla/DPO-MIX-7K
[2] https://huggingface.co/sfairXC/FsfairX-LLaMA3-RM-v0.1
[3] https://github.com/lm-sys/FastChat/pull/3158

Table 1: Overall results for DCKD and ADPA using LLaMA-3.2-1B as the student model and LLaMA-3.1-8B as the teacher. We report win rate (WR) against ADPA-trained LLaMA-3.2-1B on AlpacaEval, the average MT-Bench ratings, and Open LLM Leaderboard (OLL) scores. Best performances are highlighted in **bold**, while second-best are underlined.

| Method | DPO-MIX-7K | | | HelpSteer2 | | |
|---|---|---|---|---|---|---|
| | **MT-Bench** | **AlpacaEval WR (%)** | **OLL** | **MT-Bench** | **AlpacaEval WR (%)** | **OLL** |
| Teacher | 6.26 | 81.7 | 64.24 | 6.38 | 90.3 | 63.78 |
| Student | 3.29 | 25.1 | 42.00 | 3.29 | 36.8 | 42.00 |
| SFT | 3.34 | 35.7 | 42.00 | 3.13 | 38.7 | 41.86 |
| DPO | 3.40 | 33.2 | 42.70 | 3.38 | 39.3 | 42.47 |
| SimPO | 3.37 | 21.3 | 42.30 | 3.47 | 43.6 | 42.78 |
| WPO | 3.68 | 39.4 | 42.67 | 3.51 | 48.7 | 42.85 |
| VanillaKD | 3.40 | 34.1 | 42.53 | 3.58 | 40.2 | 41.69 |
| SeqKD | 3.74 | 29.7 | 42.17 | 3.44 | 44.4 | 41.78 |
| ATKD | 3.62 | 32.4 | 42.28 | 3.59 | 42.2 | 42.42 |
| PLaD | 3.42 | 29.3 | 42.31 | 3.36 | 37.8 | 42.50 |
| DDPO | 3.21 | 28.7 | 42.02 | 3.34 | 37.3 | 42.23 |
| DPKD | 3.29 | 28.9 | 41.87 | 3.10 | 36.5 | 41.74 |
| DCKD | 3.50 | 37.5 | 42.69 | 3.44 | 40.5 | 41.67 |
| ADPA | 3.88 | 50.0 | **43.38** | 3.62 | 50.0 | 42.60 |
| ADPA+ | **4.02** | 53.8 | 43.03 | **3.99** | 60.9 | **43.07** |

ADPA with the DCKD model and incorporates $\hat{y}$ predictions from DCKD during training (see Algorithm 2). For DPKD and PLaD, we use true preference data to ensure fairness.

**Training Details** The SFT for both student and teacher models is conducted over 3 epochs, using a learning rate of $2 \times 10^{-5}$ and a batch size of 128. The DPO teacher is trained with $\beta = 0.05$, a learning rate of $5 \times 10^{-7}$, a batch size of 128, and for 2 epochs. For DCKD and ADPA training, we employ context distillation (Bai et al., 2022) to improve efficiency. Specifically, for DCKD, we precompute and store the teacher model's top 50 probabilities for each token in the responses. For ADPA, we precompute $\log \pi_{\text{dpo}}(\cdot \mid s_t) - \log \pi_{\text{ref}}(\cdot \mid s_t)$ for the top 50 probabilities. Tokens present in the DPO teacher's top 50 but absent from the reference teacher's set have their log probabilities adjusted by subtracting the lowest probability in the reference's top 50. Conversely, tokens in the reference teacher's top 50 but absent from the DPO teacher's set are omitted. In DCKD, we experiment with $\alpha \in [0.1, 0.2, 0.5, 1, 2, 5]$, while for ADPA, we explore $\gamma \in [0.5, 1, 1.5, 2, 3, 5]$. More detailed training configuration can be found in Appendix F.

## 4.2 MAIN RESULTS

Table 1 summarizes the overall results, which emphasize the performance of LLaMA-3.2-1B due to space limitations. Additional results for other SLMs can be found in Appendix B. Several key observations can be drawn: **First**, our proposed methods, DCKD and ADPA, consistently outperform baseline approaches, demonstrating the efficacy of our dual-constrained distillation and advantage-guided techniques. For instance, when trained on DPO-MIX-7K, DCKD and ADPA achieve improvements of 0.10 and 0.48 respectively over DPO on MT-Bench. These results highlight that the preference-aligned teacher model is more effective in guiding the student to align its outputs with human preferences. **Second**, when ADPA is used as the reference in AlpacaEval, existing distillation and preference alignment methods exhibit a win rate below 50%. This shows the robustness of our approach and emphasizes the importance of integrating preference signals into the distillation process. **Third**, our proposed ADPA and ADPA+ outperform all baseline methods on the Open LLM Leaderboard, further validating their effectiveness in aligning models to handle diverse tasks. **Lastly**, initializing ADPA with a student model trained using DCKD (ADPA+) leads to improved performance compared to using either method alone. This combination allows the student model to better capture the teacher's output distribution and learn nuanced preference signals during training.

## 4.3 MODEL ABLATION

To evaluate the impact of various components in our methods, we perform ablation experiments on the DPO-MIX-7K dataset by systematically removing individual components from DCKD and ADPA. For DCKD, we substitute the DPO teacher with an SFT teacher trained on preferred responses from the preference dataset. Moreover, we examine the effect of excluding the $\mathcal{L}_{\text{KLD}-l}$ loss,

Table 2: Results of ablation for DCKD and ADPA using DPO-MIX-7K dataset. Best performances are highlighted in **bold**, while second-best are underlined.

| Method | Mistral-7B → Danube3-500M | | Mistral-7B → Danube2-1.8B | | LLaMA-2-13B → LLaMA-2-7B | |
| | AlpacaEval WR (%) | MT-Bench | AlpacaEval WR (%) | MT-Bench | AlpacaEval WR (%) | MT-Bench |
| --- | --- | --- | --- | --- | --- | --- |
| **DCKD** | **50.0** | **2.67** | **50.0** | **4.09** | **50.0** | **4.96** |
| - w/o DPO teacher | 48.2 | 2.46 | 35.6 | 3.63 | 39.1 | 4.60 |
| - w/o dispreferred response | 40.3 | 2.60 | 39.9 | 4.04 | 37.9 | 4.68 |
| **ADPA** | **50.0** | **2.56** | **50.0** | **4.12** | **50.0** | **4.53** |
| - w/o reference teacher | 31.6 | 2.43 | 36.6 | 3.78 | 46.2 | 4.45 |

which filters out dispreferred responses.[4] For ADPA, we analyze the impact of removing the reference teacher and minimizing the reverse cross-entropy between the student and the DPO teacher's output distributions. The results of these ablation studies are presented in Table 2.

These results show that removing the DPO teacher in DCKD results in noticeable performance degradation, indicating that the DPO teacher, optimized on human preference data, better aligns with human preferences and provides stronger guidance to the student model. Moreover, excluding dispreferred responses from DCKD results in significant performance drops. This result highlights the importance of dispreferred responses, which help the student model understand not only preferred behaviors but also undesirable ones, leading to better alignment with human preferences.

For ADPA, removing the reference teacher causes serious performance deterioration. For instance, in the Danube3-500M student, the MT-Bench rating drops from 2.56 to 2.43, while the AlpacaEval win rate decreases sharply from 50.0% to 31.6%. These results highlight the critical role of the advantage function, which offers essential guidance beyond the probabilities from the DPO teacher.

## 4.4 SAMPLE COMPLEXITY ANALYSIS

In reinforcement learning, *sample complexity* measures the number of interactions needed to achieve a desired performance level (Kakade, 2003). Efficient reward mechanisms with low sample complexity ensure stable training and scalability. Viewing token generation as a series of actions, a natural question arises: *How many samples are needed to identify the optimal next action under a given reward mechanism?* To answer this question, we examine the sample complexity of distribution-level advantage, token-level reward, and sequence-level reward, with detailed definitions provided in the Appendix C. While all these methods aim to align the model's output with human preferences, they differ in their level of granularity, as schematically illustrated in Figure 3.

**Theoretical Analysis** We first present a theoretical analysis of the sample complexity for these reward types. We aim to identify the optimal action $a_t^*$ at state $s_t$ based on different reward types:

*Distribution-Level Advantage*: The advantage $A^*(s_t, a_t)$ is computed directly from policy distributions without additional sampling, yielding a sample complexity of $O(1)$:

$$a_t^* = \arg\max_{a_t \in \mathcal{A}} \beta \log \left( \frac{\pi_{\text{dpo}}(a_t \mid s_t)}{\pi_{\text{ref}}(a_t \mid s_t)} \right).$$

*Token-Level Reward*: Calculating $r(s_t, a_t)$ involves enumerating all actions $a_t \in \mathcal{A}$, transitioning to $f(s_t, a_t)$, and computing rewards, incurring a sample complexity of $O(|\mathcal{A}|)$.

*Sequence-Level Reward*: Evaluating the expected cumulative reward for each action $a_t$ requires simulating future trajectories, resulting in an exponential sample complexity of $O(|\mathcal{A}|^{T-t})$:

$$a_t^* = \arg\max_{a_t \in \mathcal{A}} Q^*(s_t, a_t).$$

Therefore, the distribution-level advantage offers superior training stability and efficiency due to its lower sample complexity ($O(1)$ per action) and computational simplicity. By leveraging precomputed policy distributions without requiring additional sampling, this method minimizes variance in updates and avoids the cascading errors associated with state transitions. Its reduced dependence on costly trajectory simulations or token-level evaluations makes it scalable to larger vocabularies and contexts, supporting faster convergence and more robust alignment with human preferences.

---

[4]Excluding the $\mathcal{L}_{\text{KLD}-l}$ loss in DCKD results in a VanillaKD-like loss with minor weighting adjustments.

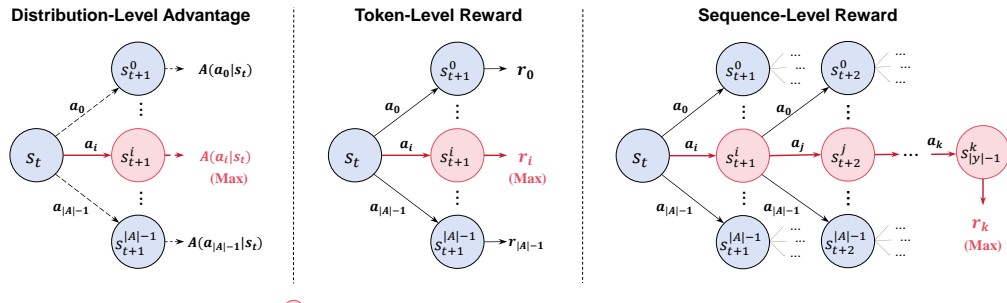

Figure 3: An illustration comparing the sample complexity of different reward granularities. Using a *distribution-level* advantage function (**left**), the model directly identifies the optimal action $a_t^* = \arg\max_{a \in \mathcal{A}} A(s_t, a)$ at state $s_t$, as shown by the red solid line, without needing to explore subsequent states or sample additional trajectories (dotted line). For *token-level* reward (**middle**), the model evaluates immediate rewards $r(s_t, a)$ for each action $a \in \mathcal{A}$, transitions to the corresponding next states $s_{t+1}$ (solid line), and may consider future rewards to determine $a_t^*$, leading to a sample complexity of $O(|\mathcal{A}|)$. With *sequence-level* reward (**right**), the model generates full trajectories starting from each possible action $a \in \mathcal{A}$, reaching EOS to receive the reward $R(\tau)$. This requires exploring all possible action sequences of length $T - t$, resulting in sample complexity $O(|\mathcal{A}|^{T-t})$.

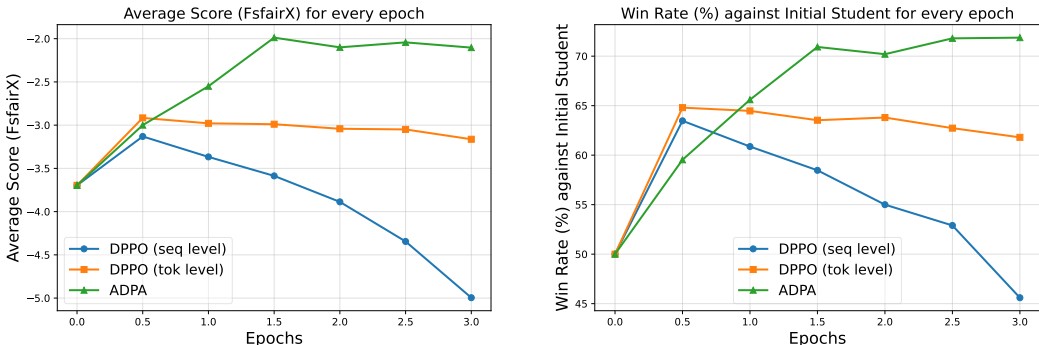

Figure 4: Comparison between ADPA and PPO-based methods on the validation set. The x-axis denotes the training epochs, and the y-axis indicates either the average scores (**left**) or the win rates (**right**) of responses generated by checkpoints during training, as evaluated using FsfairX.

**Empirical Results** To complement the theoretical analysis, we conducted experiments to empirically compare the three types of reward signals. Specifically, we compared ADPA (distribution-level advantage) with two PPO-based student models that use token-level and sequence-level rewards (distilled PPO, DPPO). Using FsfairX as the evaluator, we assessed the performance of the trained student models on the DPO-MIX-7K validation set. As shown in Figure 4, ADPA significantly stabilizes the training process compared to both token-level and sequence-level DPPO. This improvement can be attributed to two key factors: (1) ADPA offers a distribution-level preference reward signal, providing richer and more informative feedback than token- or sequence-level rewards; and (2) its offline optimization process is more stable and efficient compared to the time-consuming online reinforcement learning required by DPPO. Table 3 further presents the win rates on AlpacaEval. ADPA clearly outperforms DPPO, with token-level and sequence-level methods achieving win rates of only 40.0% and 27.7%, respectively, both well below 50%. Overall, these results demonstrate that ADPA's distribution-level advantage function provides an efficient and robust approach to preference alignment for SLMs. Its low sample complexity ensures stable training and strong performance.

## 4.5 ANALYSIS AND DISCUSSION

**Impact of $\alpha$ and $\gamma$** We further investigate the effects of the hyperparameters $\alpha$ in DCKD and $\gamma$ in ADPA on the student model's preference alignment. We report the results of distilling Mistral-7B to Danube2-1.8B on the DPO-MIX-7K dataset in Figure 5. The FsfairX reward model is employed to assess the average response scores on the validation set. To further analyze the student model's ability to learn preference information, we employed the Reward Accuracy metric (Meng et al.,

Table 3: Comparison of ADPA (distribution-level) and PPO-based DPPO with different reward granularities. The sample complexities $O(1)$, $O(|\mathcal{A}|)$, and $O(|\mathcal{A}|^{T-t})$ highlight a theoretical view of how many enumerations or simulations might be needed to find an optimal next action.

| Method | Sample Complexity | Reference | AlpacaEval WR (%) |
|---|---|---|---|
| DPPO (sequence-level) | $O(|\mathcal{A}|^{T-t})$ | ADPA | 27.7 |
| DPPO (token-level) | $O(|\mathcal{A}|)$ | ADPA | 40.0 |
| **ADPA (distribution-level)** | $\boldsymbol{O(1)}$ | ADPA | **50.0** |

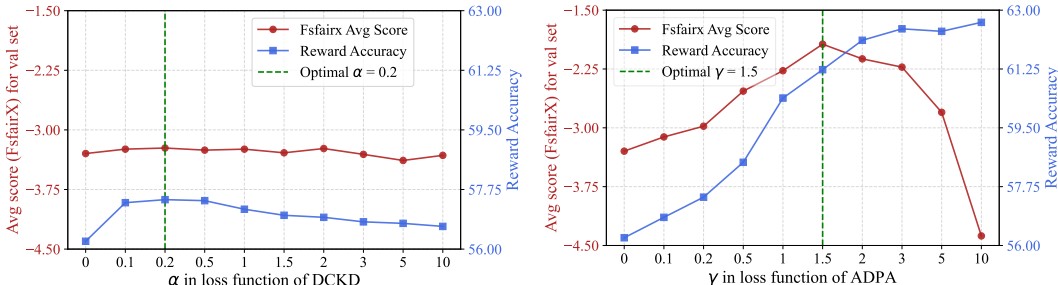

Figure 5: Impact of hyperparameters $\alpha$ in DCKD **(left)** and $\gamma$ in ADPA **(right)** on the preference alignment of Danube2-1.8B with Mistral-7B as the teacher on the DPO-MIX-7K dataset. The average scores evaluated by FsfairX and Reward Accuracy (Meng et al., 2024) are reported.

2024), which assesses the probability that the student model assigns a higher average log-probability to preferred responses compared to dispreferred ones in the preference dataset, aiming to capture the model's capability to distinguish between positive and negative samples after training.

From the left figure, it can be seen that as the value of $\alpha$ increases, the Reward Accuracy initially rises and then declines, although the changes are not particularly significant. The highest average score on the validation set, evaluated by FsfairX, is observed at $\alpha = 0.2$, indicating optimal performance at this value. From the right figure, we observe that as $\gamma$ increases, both the Reward Accuracy and the FsfairX average score consistently improve, suggesting that the student model becomes more adept at distinguishing between preferred and dispreferred responses. However, when $\gamma$ exceeds a value of 3, the model becomes over-optimized with respect to the distillation objective, leading to a decline in FsfairX scores. This indicates that an excessively large $\gamma$ causes the student model to overemphasize the advantage function signals, reducing its ability to generalize.

**Additional Analysis** We also conducted additional experiments to examine the impact of different distillation objectives based on the Q-function, with detailed results presented in Appendix D. Additionally, we analyze the influence of state source selection in ADPA, as outlined in Appendix E. Moreover, to provide a comprehensive understanding of our work and its potential improvements, we discuss limitations and future directions in Appendix G. Finally, to offer concrete examples, case studies are provided in Appendix H.

## 5 CONCLUSION

This paper explores the challenge of aligning small language models (SLMs) with human preferences, a task complicated by their limited capacity. To overcome this limitation, we proposed a teacher-student framework that leverages a well-aligned teacher model to guide the alignment process to transfer knowledge of human preferences to SLMs. Within this framework, we introduced two alignment methods: Dual-Constrained Knowledge Distillation (DCKD), which applies two KL-divergence constraints to transfer alignment knowledge, and Advantage-Guided Distillation for Preference Alignment (ADPA), which utilizes an advantage function to provide nuanced, distribution-level reward signals. Experimental results showed that both methods significantly enhance alignment in SLMs, with ADPA demonstrating particularly strong performance. Moreover, the combination of ADPA and DCKD achieved even greater alignment improvements. These results highlight the potential of utilizing larger, preference-aligned models to guide the alignment of smaller models. Future research could focus on refining the distillation process and exploring the applicability of the proposed methods across a wider range of tasks and model sizes.

ACKNOWLEDGEMENTS

This work was supported by the National Natural Science Foundation of China (No. 62176270) and the Guangdong Basic and Applied Basic Research Foundation (No. 2023A1515012832).

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

## SUMMARY OF THE APPENDIX

This appendix provides additional experimental results and further discussions related to this work.

- Appendix **A** presents **Derivation of the Q-function and Advantage Function**.
- Appendix **B** provides more **Evaluation Results for Different Student Models**.
- Appendix **C** describes more **Details of the Sequence- and Token-Level Reward**.
- Appendix **D** presents **Variants of KD Objective based on Q-function**.
- Appendix **E** provides **Impact of the Sampling Source of State in ADPA**.
- Appendix **F** gives more **Details of Training Configurations**.
- Appendix **G** adds more discussions of **Limitations and Future Work**.
- Appendix **H** includes several **Case Studies**.

---

**Algorithm 1** ADPA Training Pipeline

---

**Require:** Student model $\pi_\theta$, teacher model $\pi_{\text{tch}}$, instruction-tuning dataset $\mathcal{D}_{\text{it}}$, preference dataset $\mathcal{D}_{\text{pref}}$
**Ensure:** Trained student model $\pi_\theta''$

1: Fine-tune $\pi_{\text{tch}}$ and $\pi_\theta$ on $\mathcal{D}_{\text{it}}$ to obtain supervised fine-tuned (SFT) models for both teacher and student, named reference teacher $\pi_{\text{ref}}$ and SFT student model $\pi_\theta'$.
2: Fine-tune $\pi_{\text{ref}}$ on $\mathcal{D}_{\text{pref}}$ using DPO to obtain $\pi_{\text{dpo}}$ (DPO teacher model).
3: Create new dataset $\hat{\mathcal{D}} = \{\}$
4: **for** prompt $x$ and preferred response $y_w$ in $\mathcal{D}_{\text{pref}}$ **do**
5:     Generate outputs from the SFT student model $\pi_\theta'$ for the given prompt $x$ to obtain $\hat{y}$.
6:     $\hat{\mathcal{D}} \leftarrow \hat{\mathcal{D}} \cup \{(x, y_w, \hat{y})\}$
7: **end for**
8: Optimize the SFT student model $\pi_\theta'$ on $\hat{\mathcal{D}}$ using the ADPA loss to obtain the $\pi_\theta''$ (We utilize $y_w$ as the ground truth response within the ADPA loss framework).
9: Return the trained student model $\pi_\theta''$.

---

**Algorithm 2** ADPA+ Training Pipeline

---

**Require:** Student model $\pi_\theta$, teacher model $\pi_{\text{tch}}$, instruction-tuning dataset $\mathcal{D}_{\text{it}}$, preference dataset $\mathcal{D}_{\text{pref}}$
**Ensure:** Trained student model $\pi_\theta''$

1: Fine-tune $\pi_{\text{tch}}$ and $\pi_\theta$ on $\mathcal{D}_{\text{it}}$ to obtain supervised fine-tuned (SFT) models for both teacher and student, named reference teacher $\pi_{\text{ref}}$ and SFT student model $\pi_\theta'$.
2: Fine-tune $\pi_{\text{ref}}$ on $\mathcal{D}_{\text{pref}}$ using DPO to obtain $\pi_{\text{dpo}}$ (DPO teacher model).
3: Fine-tune $\pi_\theta'$ on $\mathcal{D}_{\text{pref}}$ by DCKD algorithm, with the guidance of $\pi_{\text{dpo}}$, to obtain $\pi_{\text{stu-DCKD}}$.
4: Create new dataset $\hat{\mathcal{D}} = \{\}$
5: **for** prompt $x$ and preferred response $y_w$ in $\mathcal{D}_{\text{pref}}$ **do**
6:     Generate outputs from DCKD student model $\pi_{\text{stu-DCKD}}$ for the given prompt $x$ to obtain $\hat{y}$.
7:     $\hat{\mathcal{D}} \leftarrow \hat{\mathcal{D}} \cup \{(x, y_w, \hat{y})\}$
8: **end for**
9: Optimize the DCKD student model $\pi_{\text{stu-DCKD}}$ on $\hat{\mathcal{D}}$ using the ADPA loss to obtain the $\pi_\theta''$ (We utilize $y_w$ as the ground truth response within the ADPA loss framework).
10: Return the trained student model $\pi_\theta''$.

---

## A  ADVANTAGE FUNCTION DERIVED FROM DPO TEACHER

We model text generation as a token-level Markov Decision Process (MDP), defined as $\mathcal{M} = (\mathcal{S}, \mathcal{A}, f, r, s_1)$, where $\mathcal{S}$ represents the state space. Each state $s_t = (x, y_{<t})$ consists of the prompt $x$ and the sequence of tokens generated up to time step $t$. The action space $\mathcal{A}$ corresponds to the vocabulary. The transition dynamics $f(s, a)$ are deterministic in text generation: the next token $a_t$ is appended to the sequence of observed text $s_t$. The initial state $s_1$ is defined by the prompt: $s_1 = x$.

A trajectory $\tau = \{(s_t, a_t)\}_{t=1}^{|\tau|}$ represents the sequence of states and actions up until the end of the generation process, with $|\tau|$ representing the length of the trajectory and $a_{|\tau|} = \text{EOS}$ (End of Sequence). The policy $\pi : \mathcal{S} \rightarrow \mathcal{A}$ outputs a probability distribution over the action space for each state. In the context of text generation, the joint probability of a trajectory $\tau$ is typically computed as the product of conditional probabilities for each token $a_t$ given its corresponding state $s_t$.

$$\pi(\tau) = \prod_{t=1}^{|\tau|} \pi(a_t \mid s_t). \tag{13}$$

The reward function $r$ assigns token-level rewards based on human preferences, and each action $a_t$ taken in state $s_t$ is associated with a reward $r(s_t, a_t)$. This reward function can be modeled using the Bradley-Terry framework (Bradley & Terry, 1952). The cumulative reward, which sums over all token-level rewards, represents the total reward for a given trajectory. The probability that the "winning" trajectory $\tau_w$ is preferred over the "losing" trajectory $\tau_l$ is defined as:

$$p^*(\tau^w > \tau^l) = \frac{\exp\left(\sum_{t=1}^{|\tau_w|} r(s_t^w, a_t^w)\right)}{\exp\left(\sum_{t=1}^{|\tau_w|} r(s_t^w, a_t^w)\right) + \exp\left(\sum_{t=1}^{|\tau_l|} r(s_t^l, a_t^l)\right)}. \tag{14}$$

Most classical RLHF approaches optimize the policy $\pi_\theta$, parameterized by $\theta$, by maximizing the cumulative reward with a penalty for the KL-divergence between the policy $\pi_\theta$ and a reference policy $\pi_{\text{ref}}$. This penalty serves to limit deviations from the reference policy. In the case of LLM generation, the discount factor is typically set to 1. The optimization objective is denoted as:

$$\max_\theta \mathbb{E}_{\pi_\theta} \left[ \sum_{t=1}^{|\tau|} \left( r(s_t, a_t) - \beta \log \frac{\pi_\theta(a_t \mid s_t)}{\pi_{\text{ref}}(a_t \mid s_t)} \right) \right].$$

Then, we define the value function $V(s_t)$ as the expected total reward that the policy $\pi_\theta$ can obtain in the future, starting from a given state $s_t$:

$$V(s_t) = \mathbb{E}_{\pi_\theta} \left[ \sum_{i=1}^{|\tau|-t} \left( r(s_{t+i}, a_{t+i}) - \beta \log \frac{\pi_\theta(a_{t+i} \mid s_{t+i})}{\pi_{\text{ref}}(a_{t+i} \mid s_{t+i})} \right) \right]. \tag{15}$$

The action-value function $Q(s_t, a_t)$ denotes the expected cumulative reward over future time steps, obtained by taking action $a_t$ in state $s_t$ and subsequently adhering to the policy $\pi_\theta$. This is formalized by the Bellman equation as:

$$Q(s_t, a_t) = r(s_t, a_t) + V(s_{t+1}), \tag{16}$$

where $s_{t+1} = f(s_t, a_t)$ is the next state.

The value function $V(s_t)$ can be denoted as an expectation over $Q(s_t, a_t)$ under the policy $\pi_\theta$ with the KL penalty:

$$V(s_t) = \sum_{a \in \mathcal{A}} \pi_\theta(a_t \mid s_t) \left[ Q(s_t, a_t) - \beta \log \frac{\pi_\theta(a_t \mid s_t)}{\pi_{\text{ref}}(a_t \mid s_t)} \right]. \tag{17}$$

To maximize this value function under the constraint $\sum_{a_t \in \mathcal{A}} \pi_\theta(a_t \mid s_t) = 1$, we employ the Lagrange multiplier method. Let $\lambda$ be the Lagrange multiplier, the objective function is defined as:

$$\mathcal{L} = \sum_{a \in \mathcal{A}} \pi_\theta(a_t \mid s_t) \left[ Q(s_t, a_t) - \beta \log \frac{\pi_\theta(a_t \mid s_t)}{\pi_{\text{ref}}(a_t \mid s_t)} \right] - \lambda \left( \sum_{a_t \in \mathcal{A}} \pi_\theta(a_t \mid s_t) - 1 \right). \tag{18}$$

Solving this function by setting $\frac{\partial(\mathcal{L})}{\partial \pi_\theta(a_t|s_t)} = 0$ and $\frac{\partial(\mathcal{L})}{\partial \lambda} = 0$, we obtain the optimal policy $\pi^*$:

$$\pi^*(a_t \mid s_t) = \frac{\pi_{\text{ref}}(a_t \mid s_t) \exp\left(Q^*(s_t, a_t)/\beta\right)}{\sum_{a \in \mathcal{A}} \pi_{\text{ref}}(a_t \mid s_t) \exp\left(Q^*(s_t, a_t)/\beta\right)}. \tag{19}$$

Substituting $\pi_\theta = \pi^*$ into Eq. (17), the corresponding optimal value function $V^*(s_t)$ and the simplified expression for $\pi^*$ can be represented as:

$$V^*(s_t) = \beta \log \sum_{a \in \mathcal{A}} \pi_{\text{ref}}(a_t \mid s_t) \exp\left(Q^*(s_t, a_t)/\beta\right), \tag{20}$$

$$\pi^*(a_t \mid s_t) = \pi_{\text{ref}}(a_t \mid s_t) \exp\left(\frac{Q^*(s_t, a_t) - V^*(s_t)}{\beta}\right). \tag{21}$$

Here, $Q^*(s_t, a_t)$ and $V^*(s_t)$ represent the optimal Q-function and value function for $\pi_\theta = \pi^*$. From Eq. (21), we have the following equality:

$$Q^*(s_t, a_t) - V^*(s_t) = \beta \log \frac{\pi^*(a_t \mid s_t)}{\pi_{\text{ref}}(a_t \mid s_t)}. \tag{22}$$

Next, since no future reward can be received after the final time step, we have the equation $V^*(s_{|\tau|+1}) = 0$. Applying the Bellman equation for the reward $r(s_t, a_t) = Q^*(s_t, a_t) - V^*(s_{t+1})$, we can denote the sum of rewards for trajectory $\tau$ in terms of the optimal policy $\pi^*$ as follows:

$$\sum_{t=1}^{|\tau|} r(s_t, a_t)$$

$$= \sum_{t=1}^{|\tau|} [Q^*(s_t, a_t) - V^*(s_{t+1})]$$

$$= \sum_{t=2}^{|\tau|} [Q^*(s_t, a_t) - V^*(s_t)] + Q^*(s_1, a_1) - V^*(s_{|\tau|+1})$$

$$= \sum_{t=2}^{|\tau|} [Q^*(s_t, a_t) - V^*(s_t)] + \left(\beta \log \frac{\pi^*(a_1 \mid s_1)}{\pi_{\text{ref}}(a_1 \mid s_1)} + V^*(s_1)\right) - 0$$

$$= \sum_{t=1}^{|\tau|} \beta \log \frac{\pi^*(a_t \mid s_t)}{\pi_{\text{ref}}(a_t \mid s_t)} + V^*(s_1). \tag{23}$$

Substituting the sum of rewards from Eq. (23) into the token-level Bradley-Terry model in Eq. (14), we derive the Maximum Likelihood Estimation (MLE) objective for the log-likelihood as follows:

$$\pi^* = \arg\max_\theta \mathbb{E}_{(\tau_w, \tau_l) \sim \mathcal{D}} \left[\log \sigma \left(\beta \sum_{i=1}^{|\tau_w|} \log \frac{\pi_\theta(a_i^w \mid s_i^w)}{\pi_{\text{ref}}(a_i^w \mid s_i^w)} - \beta \sum_{i=1}^{|\tau_l|} \log \frac{\pi_\theta(a_i^l \mid s_i^l)}{\pi_{\text{ref}}(a_i^l \mid s_i^l)}\right)\right]$$

$$= \arg\min_\theta -\mathbb{E}_{(\tau_w, \tau_l) \sim \mathcal{D}} \left[\log \sigma \left(\beta \log \frac{\pi_\theta(\tau_w)}{\pi_{\text{ref}}(\tau_w)} - \beta \log \frac{\pi_\theta(\tau_l)}{\pi_{\text{ref}}(\tau_l)}\right)\right]. \tag{24}$$

Thus, we conclude that the optimal policy derived from token-level RLHF can minimize the DPO objective, and both objectives yield the same optimal policy.

The advantage function denotes the additional benefit of taking an action $a_t$ in state $s_t$ relative to the expected return of taking the average action. From Eq. (22), we obtain the advantage function as:

$$A^*(a_t, s_t) = Q^*(s_t, a_t) - V^*(s_t) = \beta \log \pi^*(a_t \mid s_t) - \beta \log \pi_{\text{ref}}(a_t \mid s_t). \tag{25}$$

Since the objectives of DPO and token-level RLHF yield the same optimal policy, we substitute $\pi^*$ with $\pi_{\text{dpo}}$, and define the advantage function derived from $\pi_{\text{dpo}}$ and $\pi_{\text{ref}}$ as follows:

$$A_{\text{dpo}}(a_t, s_t) = \beta \log \frac{\pi_{\text{dpo}}(a_t \mid s_t)}{\pi_{\text{ref}}(a_t \mid s_t)}.$$

# B   MORE EVALUATION RESULTS ON SLMS

We present the MT-Bench ratings and AlpacaEval win rates (WR) for student models, including Danube3-500M, Danube2-1.8B, and LLaMA-2-7B, trained on the DPO-MIX-7K and HelpSteer2 datasets, in Table 4. Moreover, we provide the evaluation results of LLaMA-3.2-1B across various benchmarks on the OpenLLM leaderboard in Table 5.

From Table 4, we note that employing ADPA as the reference model leads to a win rate consistently lower than 50.0% for most models on AlpacaEval. This highlights the effectiveness of ADPA in aligning models with human preferences, as its outputs are generally favored over those generated by other methods. The combination of ADPA with DCKD, forming ADPA+, yields even further improvements. For instance, when using the Danube2-1.8B model on the DPO-MIX-7K dataset, ADPA+ achieves an MT-Bench score of 4.40, significantly outperforming DCKD (4.09) and DPO (3.87). When using HelpSteer2 as the training set, ADPA+ achieves a 62.7% win rate in the AlpacaEval evaluation, marking the highest improvement across all methods.

For smaller models like Danube3-500M, ADPA+ achieves substantial improvements, which is particularly important for resource-constrained applications. Compared to DPKD, ADPA+ enhances the MT-Bench rating for Danube3-500M from 2.66 to 2.75 using the DPO-MIX-7K dataset and increases the AlpacaEval win rate by 16.9% (from 36.3% to 53.2%) using HelpSteer2, showcasing its effectiveness in boosting the performance of small language models (SLMs).

For larger student models like LLaMA-2-7B, ADPA+ delivers significant improvements. On the DPO-MIX-7K training dataset, ADPA+ attains an MT-Bench score of 5.08, outperforming DCKD (4.96), DPO (4.29), and ADPA (4.53). This underscores the advantages of DCKD initialization and the enhanced preference signals ADPA+ provides for larger language models.

From Table 5, we observe that ADPA+ demonstrates remarkable performance on the OpenLLM leaderboard, achieving state-of-the-art results on critical benchmarks including ARC (42.15) and HellaSwag (70.49) when trained on DPO-MIX-7K. In addition, when trained on the HelpSteer2 dataset, ADPA+ attains an impressive average score of 43.07, surpassing all baseline methods such as WPO and PLaD by a significant margin. These results collectively substantiate ADPA+'s exceptional capability in aligning models with human preferences while simultaneously preserving robust generalization across diverse benchmark tasks.

These results show that ADPA+ not only improves alignment performance but also enhances general task performance, making it a versatile and effective approach across both smaller and larger models.

# C   DETAILS OF THE SEQUENCE- AND TOKEN-LEVEL REWARDS

This section provides a detailed elaboration on the sequence-level and token-level rewards discussed in Section 4.4.

**Sequence-Level Reward**   The sequence-level reward, $r_{\text{seq-level}}$, is defined using the DPO teacher $\pi_{\text{dpo}}$ and the reference teacher $\pi_{\text{ref}}$ as follows:

$$r_{\text{seq-level}}(x, y) = \beta \log \frac{\pi_{\text{dpo}}(y \mid x)}{\pi_{\text{ref}}(y \mid x)} = \beta \sum_{t=1}^{|y|} \log \frac{\pi_{\text{dpo}}(y_t \mid x, y_{<t})}{\pi_{\text{ref}}(y_t \mid x, y_{<t})}. \tag{26}$$

Here, $\beta$ is a hyperparameter used during the training of the DPO teacher. The reward is assigned to the end of sequence (EOS), while all positions are regulated by a KLD penalty. Let $\pi_{\text{S-ref}}$ denote the reference model, which is initialized from the student model after supervised fine-tuning on the instruction-tuning dataset. The sequence-level reward with a KL penalty, $r_{\text{seq-level w/ KL penalty}}$, for each token at time step $t$ is represented as:

$$r_{\text{seq-level w/ KL penalty}}(x, y, y_t) = \begin{cases} 0 - \beta \log \frac{\pi_\theta(y_t \mid x, y_{<t})}{\pi_{\text{S-ref}}(y_t \mid x, y_{<t})}, & \text{if } y_t \neq \text{EOS}, \\ r_{\text{seq-level}}(x, y) - \beta \log \frac{\pi_\theta(y_t \mid x, y_{<t})}{\pi_{\text{S-ref}}(y_t \mid x, y_{<t})}, & \text{if } y_t = \text{EOS}. \end{cases} \tag{27}$$

**Token-Level Reward**   Each token in the sequence, whether located at the end of sequence or not, receives an individual token-level reward, $r_{\text{token-level}}$, from the DPO teacher and the reference teacher,

Table 4: Overall results of our methods using Danube3-500M, Danube2-1.8B, and LLaMA-2-7B as the student models. We show the win rate (WR) against ADPA-trained student models on AlpacaEval , and the average rating on MT-Bench. Best performances are highlighted in **bold**, while second-based are underlined.

| Model | Method | DPO-MIX-7K | | HelpSteer2 | |
|---|---|---|---|---|---|
| | | AlpacaEval WR (%) | MT-Bench | AlpacaEval WR (%) | MT-Bench |
| | Teacher | 85.2 | 5.64 | 93.9 | 5.59 |
| | Student | 34.4 | 2.54 | 38.0 | 2.54 |
| | SFT | 37.1 | 2.49 | 32.4 | 2.29 |
| | DPO | 35.1 | 2.52 | 36.1 | 2.52 |
| | VanillaKD | 37.1 | 2.53 | 36.2 | 2.28 |
| Danube3-500M | SeqKD | 39.4 | 2.44 | 41.7 | 2.46 |
| | ATKD | 38.0 | 2.58 | 35.5 | 2.50 |
| | PLaD | 35.1 | 2.51 | 38.0 | 2.58 |
| | DDPO | 37.3 | 2.55 | 37.0 | 2.58 |
| | DPKD | 34.3 | 2.66 | 36.3 | 2.51 |
| | DCKD | 38.9 | 2.67 | 34.2 | 2.60 |
| | ADPA | **50.0** | 2.56 | 50.0 | 2.70 |
| | ADPA+ | 49.0 | **2.75** | **53.2** | **2.76** |
| | Teacher | 61.1 | 5.64 | 82.5 | 5.59 |
| | Student | 28.6 | 3.80 | 39.5 | 3.80 |
| | SFT | 29.1 | 3.97 | 40.4 | 3.89 |
| | DPO | 31.4 | 3.87 | 40.3 | 3.90 |
| | VanillaKD | 28.3 | 3.98 | 46.3 | 4.03 |
| Danube2-1.8B | SeqKD | 32.8 | 3.94 | 42.3 | 4.10 |
| | ATKD | 29.8 | 3.84 | 42.9 | 3.93 |
| | PLaD | 29.1 | 3.92 | 44.4 | 3.84 |
| | DDPO | 31.7 | 3.82 | 39.2 | 3.68 |
| | DPKD | 38.7 | 4.34 | 43.2 | 3.97 |
| | DCKD | 34.2 | 4.09 | 51.1 | 4.05 |
| | ADPA | 50.0 | 4.12 | 50.0 | 4.04 |
| | ADPA+ | **61.0** | **4.40** | **62.7** | **4.33** |
| | Teacher | 42.6 | 5.65 | 71.3 | 5.43 |
| | Student | 21.5 | 4.26 | 24.0 | 4.26 |
| | SFT | 21.6 | 4.70 | 35.7 | 4.30 |
| | DPO | 28.7 | 4.29 | 38.6 | 4.51 |
| | VanillaKD | 29.5 | 4.72 | 35.3 | 4.60 |
| LLaMA-2-7B | SeqKD | 25.0 | 4.67 | 28.6 | 4.47 |
| | ATKD | 24.1 | 4.56 | 32.0 | 4.43 |
| | PLaD | 21.7 | 4.20 | 28.0 | 4.35 |
| | DDPO | 21.7 | 4.31 | 30.4 | 3.78 |
| | DPKD | 22.3 | 4.28 | 28.7 | 3.97 |
| | DCKD | 32.5 | 4.96 | 38.3 | 4.41 |
| | ADPA | 50.0 | 4.53 | 50.0 | 4.40 |
| | ADPA+ | **59.6** | **5.08** | **60.1** | **4.86** |

defined as:

$$r_{\text{token-level}}(\{x, y_{<t}\}, y_t) = \beta \log \frac{\pi_{\text{dpo}}(y_t \mid x, y_{<t})}{\pi_{\text{ref}}(y_t \mid x, y_{<t})}. \tag{28}$$

In our experiments, the token-level reward is computed as the difference in the output log probabilities between the DPO teacher and the reference teacher (Zhong et al., 2024a). The token-level reward with a KL penalty, $r_{\text{token-level w/KL penalty}}$, for each token at time step $t$ is then given by:

$$r_{\text{token-level w/KL penalty}}(x, y_{<t}, y_t) = r_{\text{token-level}}(\{x, y_{<t}\}, y_t) - \beta \log \frac{\pi_\theta(y_t \mid x, y_{<t})}{\pi_{\text{S-ref}}(y_t \mid x, y_{<t})}. \tag{29}$$

The token-level and sequence-level rewards with KL penalty, as defined above, are used to optimize the student model using PPO, referred to as Distilled PPO (DPPO). To ensure a fair comparison between ADPA and DPPO with different reward formulations and to enhance the stability of the online RL process in DPPO, we incorporate $\mathcal{L}_{\text{SFT}}$ with a weight of 1 into the overall loss[5].

---

[5]The overall loss also includes the policy loss and critic loss, as is standard in PPO training.

Table 5: Evaluation results across different datasets and training methods for LLaMA-3.2-1B. Benchmarks include AI2 Reasoning Challenge (ARC), HellaSwag, TruthfulQA, MMLU, Winogrande, GSM8K, and the overall average. Best performances are highlighted in **bold**, while second-based are underlined.

| Training Dataset | Method | ARC | HellaSwag | TruthfulQA | MMLU | Winogrande | GSM8K | Average |
|---|---|---|---|---|---|---|---|---|
| DPO-MIX-7K | Teacher | 57.17 | 83.60 | 57.78 | 63.16 | 75.69 | 48.01 | 64.24 |
| | Student | 41.04 | 68.38 | 40.91 | 34.38 | 60.46 | 6.82 | 42.00 |
| | SFT | 41.04 | 68.38 | 40.91 | 34.38 | 60.46 | 6.82 | 42.00 |
| | DPO | 40.27 | 68.51 | 45.17 | 34.38 | 61.25 | 6.59 | 42.70 |
| | SimPO | 39.85 | 68.06 | 44.04 | 34.26 | 61.72 | 5.84 | 42.30 |
| | WPO | 41.21 | 69.30 | 45.19 | 33.57 | 61.01 | 5.76 | 42.67 |
| | VanillaKD | 41.04 | 67.25 | 42.91 | 34.04 | **62.90** | 7.05 | 42.53 |
| | SeqKD | 40.02 | 66.49 | 44.11 | **35.28** | 61.25 | 5.84 | 42.17 |
| | ATKD | 40.27 | 67.33 | 44.65 | 34.38 | 60.77 | 6.29 | 42.28 |
| | PLaD | 39.93 | 68.22 | 44.55 | 34.42 | 61.56 | 5.16 | 42.31 |
| | DDPO | 39.51 | 66.83 | 44.55 | 33.91 | 61.72 | 5.61 | 42.02 |
| | DPKD | 38.74 | 66.18 | 45.18 | 34.57 | 61.32 | 5.23 | 41.87 |
| | DCKD | 41.21 | 67.64 | 43.38 | 34.57 | 61.80 | **7.51** | 42.69 |
| | ADPA | 41.81 | 68.66 | **47.98** | 34.74 | 61.80 | 5.31 | **43.38** |
| | ADPA+ | **42.15** | **70.49** | 45.89 | 34.60 | 59.91 | 5.16 | 43.03 |
| HelpSteer2 | Teacher | 56.48 | 83.24 | 57.33 | 63.29 | 75.77 | 46.55 | 63.78 |
| | SFT | 41.81 | 67.22 | 43.90 | 30.78 | 60.62 | **6.82** | 41.86 |
| | DPO | 39.42 | 68.15 | 45.27 | 34.27 | 62.04 | 5.69 | 42.47 |
| | SimPO | 39.68 | 68.41 | 45.72 | 34.44 | 62.43 | 5.99 | 42.78 |
| | WPO | 39.51 | 68.60 | 46.22 | 34.38 | 62.27 | 6.14 | 42.85 |
| | VanillaKD | 40.36 | 67.14 | 44.65 | 29.83 | 61.80 | 6.37 | 41.69 |
| | SeqKD | 40.61 | 65.78 | 43.46 | 33.81 | 61.25 | 5.77 | 41.78 |
| | ATKD | 40.10 | 67.38 | 44.64 | 34.56 | 61.40 | 6.44 | 42.42 |
| | PLaD | 40.19 | 68.05 | 44.76 | 34.37 | 62.12 | 5.53 | 42.50 |
| | DDPO | 38.74 | 66.19 | 46.22 | 33.64 | **62.59** | 5.99 | 42.23 |
| | DPKD | 38.74 | 65.87 | 45.00 | 33.55 | 61.96 | 5.31 | 41.74 |
| | DCKD | 40.36 | 66.95 | 45.31 | 29.95 | 61.48 | 5.99 | 41.67 |
| | ADPA | 42.24 | **69.24** | 45.46 | 31.53 | 61.25 | 5.91 | 42.61 |
| | ADPA+ | **42.49** | 69.12 | **46.94** | **35.20** | 60.14 | 4.54 | **43.07** |

# D  VARIANTS OF KD OBJECTIVE BASED ON Q-FUNCTION

In this section, we explore alternative ways of leveraging the Q-function derived from the DPO teacher and reference teacher for KD. By comparing ADPA with these Q-function-based knowledge distillation (KD) variants, we aim to provide a more comprehensive understanding of whether different advantage-based KD objectives can further improve the student's alignment performance.

Specifically, we apply the *argmax* or *softmax* operation on $A_{\text{dpo}}(\cdot \mid s)$ to obtain teacher policies[6], and then perform knowledge distillation by minimizing the cross-entropy (CE) loss or Kullback-Leibler Divergence (KLD) between the student policy and the teacher policies deduced from the advantage function (Rusu et al., 2015; Czarnecki et al., 2019).

Table 6: Results of Q-argmax KD and Q-softmax KD against ADPA on AlpacaEval.

| Method | Reference | WR (%) |
|---|---|---|
| Q-argmax KD | ADPA | 41.8 |
| Q-softmax KD | ADPA | 28.2 |
| **ADPA** | ADPA | **50.0** |

**Q-argmax KD:**  The Q-argmax KD method applies *argmax* to $A_{\text{dpo}}(\cdot \mid s)$, producing a one-hot distribution that assigns a value of 1 to the action with the highest advantage. Through distillation, this approach allows the student model to focus on replicating the most confident decisions of the advantage function. Let $\hat{D}$ denotes the dataset containing prompts $x$, ground truth responses $y$, and

---

[6]Adding constants to the inputs of softmax and argmax does not affect the results. Therefore, we apply these operations to the advantage function in Eq. (8) to obtain the policy derived from Q-function.

student's generated responses $\hat{y}$. The overall Q-argmax KD loss function is defined as follows:

$$\mathcal{L}_{\text{Q-argmax}} = \mathbb{E}_{(x,y,\hat{y})\sim\hat{\mathcal{D}}} \left[ \mathcal{L}_{\text{SFT}}(x,y) + \frac{\gamma}{|\hat{y}|} \sum_{t=1}^{|\hat{y}|} \text{CE}\left( \mathbf{1}\left\{ \arg\max_{a_t\in\mathcal{A}}(A_{\text{dpo}}(s_t,a_t)) \right\}, \pi_\theta(\cdot \mid s_t) \right) \right].$$

(30)

**Q-softmax KD:** Applying the *softmax* function to $A_{\text{dpo}}(\cdot \mid s)$ before distillation enables the student model to learn from the full policy distribution of the advantage function. This approach captures subtle nuances in decision-making, extending beyond merely selecting the action with the highest Q-value. Similar to Q-argmax KD, the overall Q-softmax KD loss function is formulated as follows:

$$\mathcal{L}_{\text{Q-softmax}} = \mathbb{E}_{(x,y,\hat{y})\sim\hat{\mathcal{D}}} \left[ \mathcal{L}_{\text{SFT}}(x,y) + \frac{\gamma}{|\hat{y}|} \sum_{t=1}^{|\hat{y}|} D_{\text{KL}}\left( \text{softmax}(A_{\text{dpo}}(s_t,\cdot)) \parallel \pi_\theta(\cdot \mid s_t) \right) \right].$$

(31)

We evaluated Q-argmax KD and Q-softmax KD against ADPA using the DPO-MIX-7K dataset with the Danube2-1.8B model, with results shown in Table 6. Compared to ADPA as the reference, Q-argmax KD achieves a win rate of 41.8%, and Q-softmax KD trails at 28.2%, both below ADPA's 50.0% self-comparison rate. Q-argmax KD applies an argmax operation to $A_{\text{dpo}}(\cdot \mid s)$, producing a one-hot distribution that emphasizes the teacher's top action but sacrifices nuance, limiting generalization as reflected in its 41.8% win rate. Q-softmax KD's softmax approach over-softens the distribution, introducing noise and reducing alignment, resulting in a 28.2% win rate. In contrast, ADPA leverages $A_{\text{dpo}}(\cdot \mid s)$ directly, preserving the teacher's preferences without transformation.

# E    IMPACT OF THE SAMPLING SOURCES OF STATE $s_t$ IN ADPA

In the optimization objective of Advantage-Guided Distillation for Preference Alignment (ADPA), as defined in Equation (9), the state $s_t$ comprises the prompt $x$ and the response $\hat{y}_{<t}$ generated by the student model up to time step $t$. This design implies that the student-generated text serves as the sampling source for $s_t$. To thoroughly understand the implications of this choice, we explore alternative sampling sources for $s_t$, extending beyond the student model's own responses. Specifically, we evaluate three distinct alternatives: (1) utilizing preferred responses from the preference dataset as $\hat{y}$, (2) employing dispreferred responses from the same dataset, and (3) using text generated by the teacher model as $\hat{y}$.

We tested these alternatives using the DPO-MIX-7K dataset with the Danube2-1.8B model, comparing their performance to the standard ADPA approach—where $s_t$ is sampled from the student's own responses—using the win rate (WR) on AlpacaEval. The standard ADPA configuration serves as the baseline. The comparative results, presented in Table 7, reveal significant performance variations: sampling from preferred responses yielded a WR of 30.6%, dispreferred responses achieved 49.1%, and teacher-generated responses resulted in 30.5% WR.

The superior performance of the default ADPA method can be attributed to its alignment between the training and inference environments. By sampling $s_t$ from the student's own generated responses, the training process mirrors the conditions the model will encounter during inference. This consistency ensures that the student model is optimized in a context directly reflective of its operational setting, enhancing its ability to generalize and perform effectively in real-world applications. Conversely, using external sources—such as preferred or dispreferred responses from the dataset or

Table 7: Comparison of different sources of $s_t$ in Eq. (9) against ADPA on AlpacaEval.

| Method | Reference | WR (%) |
|---|---|---|
| $s_t$ from preferred responses | ADPA | 30.6 |
| $s_t$ from dispreferred responses | ADPA | 49.1 |
| $s_t$ from teacher generated responses | ADPA | 30.5 |
| $s_t$ from student generated responses (default in ADPA) | ADPA | **50.0** |

teacher-generated text—introduces a mismatch between training and inference environments. This discrepancy can impair the student model's capacity to accurately learn and internalize nuanced preference signals, as the states it encounters during training do not correspond to those it will generate during deployment.

## F  DETAILS OF TRAINING CONFIGURATIONS

In our experiments, we train teacher models (LLaMA-3.1-8B, Mistral-7B and LLaMA-2-13B) and LLaMA-2-7B student models on a single node equipped with 8 NVIDIA A800 GPUs. For smaller student models (LLaMA-3.2-1B, H2O-Danube2-1.8B and H2O-Danube3-500M), we use a single node with 4 NVIDIA RTX 3090 GPUs. All experiments are optimized using the AdamW optimizer (Loshchilov & Hutter, 2019) with parameters $\beta_1 = 0.9$ and $\beta_2 = 0.999$, along with a weight decay of 0.0 and gradient clipping set to 1.0. We employ a cosine learning rate schedule with a maximum learning rate of $1.0 \times 10^{-5}$ and a warmup ratio of 0.1. Our global batch size is set as 128 across SFT, DPO, VanillaKD, SeqKD, DCKD, ADPA, and ADPA+. Hyperparameters for other baseline methods are set according to the recommendations in their respective papers. The training framework is built upon the Hugging Face Transformers library (Wolf et al., 2020) and the Alignment Handbook (Tunstall et al., 2024).

## G  LIMITATIONS AND FUTURE WORK

**Limitations**  While our proposed methods, DCKD and ADPA, demonstrate substantial improvements in aligning SLMs with human preferences, several limitations deserve consideration:

- **Dependence on Teacher Models**: The effectiveness of our approaches relies heavily on the availability of well-aligned teacher models. If such teacher models are not accessible or are misaligned, the performance gains of the student models may be limited.

- **Computational Overhead**: Calculating the advantage function $A_{\text{dpo}}$ in ADPA requires retrieving output probabilities from both the DPO teacher and the reference teacher at each token generation step. The inclusion of an additional teacher model introduces extra computational burden compared to single-teacher setups. However, this overhead is effectively mitigated by precomputing the log probabilities of the vocabulary for each token in the student-generated text using the DPO teacher and the reference teacher separately, followed by their subtraction to derive $A_{\text{dpo}}$. Once precomputation is completed, the training cost of ADPA aligns closely with that of other distillation methods, such as VanillaKD.

- **Generalization across Domains**: Our experiments are primarily conducted on specific datasets and domains. The generalizability of our methods to a broader range of tasks, languages, and more diverse datasets remains to be thoroughly investigated.

**Future Work**  Building upon our findings, several avenues for future research are identified:

- **Enhancing Teacher Model Alignment**: Investigate strategies for improving the alignment of teacher models when well-aligned teachers are not readily available. This could involve developing more effective alignment techniques for large models or utilizing ensemble methods.

- **Reducing Computational Complexity**: Explore techniques to minimize the computational overhead associated with computing the advantage function. Potential solutions include approximation methods, or leveraging parallel computation.

- **Cross-Domain Generalization**: Evaluate the effectiveness of the proposed methods across a wider array of tasks, languages, and domains to assess their robustness and adaptability in different settings.

## H  CASE STUDIES

We provide several cases generated by the models distilled by different methods based on the Danube2-1.8B model in Table 8. The prompts are sampled from the validation set of DPO-MIX-7K. We find that ADPA+ generates more accurate responses compared with baselines.

Table 8: Case study of a preference alignment task from the validation set of DPO-MIX-7K. Our ADPA+ model generates more accurate and concise responses, demonstrating superior alignment with human preferences compared to baseline methods like SFT and DPO.

| Task Type: Instruction Following |
|---|
| **Instruction:** 
 The task is to read a passage and convert numbers in their text forms into digit forms. The text numbers can be used to represent Quantity, Dates and Time, Addresses, or Positions. 
 One example is below: 
 Q: Jacob ate three burgers today. 
 A: Jacob ate 3 burgers today. 
 Rationale: The number in text form "three" is converted into the number in digit form "3". 
 Q: If Oswald left the Depository Building at twelve:thirty three p.m., 
 A: |
| **Reference Answer (Teacher Model):** 
 If Oswald left the Depository Building at 12:33 p.m. 
 Rationale: The text "twelve:thirty three p.m." is converted to "12:33 p.m." in digit form. |
| **SFT w/o KD Answer:** 
 If Oswald left the Depository Building at 12:33 p.m. 
 Rationale: The number in text form "12:33 p.m." is converted into the number in digit form "1233". |
| **DPO Answer:** 
 If Oswald left the Depository Building at 12:33 p.m., 
 To solve this task, I will follow these steps: 
 1. Identify the text numbers in the given passage. 
 2. Convert each text number into its corresponding digit form. 
 3. Combine the digit forms to represent the numbers in the text. 
 Here's the step-by-step process: 
 1. Identify the text numbers in the given passage: 
 * Jacob ate three burgers today. 
 * If Oswald left the Depository Building at twelve:thirty three p.m., 
 2. Convert each text number into its corresponding digit form: 
 * Jacob ate 3 burgers today. (3 in decimal) 
 * If Oswald left the Depository Building at twelve:thirty three p.m., (12 in decimal, 30 in decimal, 33 in decimal) 
 3. Combine the digit forms to represent the numbers in the text: 
 * Jacob ate 3 burgers today. (3 in decimal) 
 * If Oswald left the Depository Building at twelve:thirty three p.m., (12 in decimal, 30 in decimal, 33 in decimal) 
 So, the final answer is: 
 * Jacob ate 3 burgers today. 
 * If Oswald left the Depository Building at twelve:thirty three p.m., |
| **DCKD Answer:** 
 If Oswald left the Depository Building at 12:33 p.m. 
 In this example, the text number "twelve:thirty three" is converted into the digit form "12:33". |
| **ADPA Answer:** 
 If Oswald left the Depository Building at 12:33 p.m. |
| **ADPA+ Answer:** 
 If Oswald left the Depository Building at 12:33 p.m. |
| **ADPA w/o Reference Teacher Answer:** 
 If Oswald left the Depository Building at 12:33 p.m. 
 Rationale: The number in text form "12:33 p.m." is converted into the number in digit form "1233". |

