# OpenReview forum: "Advantage-Guided Distillation for Preference Alignment in Small Language Models"
_ICLR.cc/2025/Conference — ICLR 2025 Spotlight_

### Official Review · Reviewer_3nf4 · 2024-10-15

**Soundness:** 3
**Presentation:** 2
**Contribution:** 2
**Rating:** 6
**Confidence:** 3

**Summary:**

This paper explores knowledge distillation (KD) for LLMs during the preference alignment stage. It first introduces a simple baseline DCKD which applies Vanilla KLD-based KD on both the positive and negative examples. Then, the paper propose ADPA to enhance the contrastive signals of KD. Experiments show the effectiveness of ADPA and its components, suggesting the importance of leveraging the perference signals for KD.

**Strengths:**

1. How to leverage the preference signals for KD is an important but under-explored problem.
2. The methods are concise and easy to implement.
3.  The empirical results of ADPA is strong.

**Weaknesses:**

1. Although ADPA seems effective empirically, it is still unclear how the improvement is related to the motivation of the method. From Section 3.3 and Algorithm 1, it seems ADPA does not need the preference labels: only the prompts and the grouth truth reponses works for ADPA. How is ADPA related to preference alignment?
2. From Table 2, it seems that the reference teacher model is critical for the effectiveness of ADPA. It would be better to add more explanation on why the difference between $\pi_{dpo}$ and $\pi_{ref}$ should be considered, rather than $\pi_{dpo}$. Furthermore, what if $\pi_{dpo}$ is removed from Equation (11)? (to show the effect of $-\log \pi_{sft}$ alone)

**Questions:**

See Weaknesses.

---

> ### Author Response · Authors · 2024-11-21
> **Response to Reviewer 3nf4**
>
> Thank you for your thoughtful review and for recognizing the importance of our contribution to leveraging preference signals in knowledge distillation.
>
> **W1: Unclear how the improvement is related to the motivation of the method; ADPA seems not to need preference labels during training; how is ADPA related to preference alignment?**
>
> **Response:**
> Thank you for the insightful question. While ADPA does not directly use preference labels during the student model's training phase, it crucially relies on the advantage function: $A\_{\text{dpo}}(s\_t, a\_t) = \beta \left( \log \pi\_{\text{dpo}}(a\_t|s\_t) - \log \pi\_{\text{ref}}(a\_t|s\_t) \right),
> $ which requires log probabilities from both the DPO teacher and the reference teacher.
>
> The DPO teacher is trained using preference data, and thus, the advantage function inherently captures the preference information learned by the teacher. By distilling this advantage function, the student model learns to prioritize actions that are preferred according to the teacher's alignment with human preferences.
>
> In essence, ADPA transfers the preference alignment from the teacher to the student, even though the student does not directly use the preference labels during its own training. We have clarified this relationship in the revised paper, emphasizing how the advantage function serves as a channel for preference information.
>
> ---
>
> **W2: Why consider the difference between $\pi\_{\text{dpo}} $ and $\pi\_{\text{ref}} $ instead of $\pi\_{\text{dpo}} $ alone? What happens if $\pi\_{\text{dpo}} $ is removed from Equation (11)?**
>
> **Response:**
> Thank you for raising this important point and for providing us with the opportunity to explain our design choice in detail. The reference model $\pi\_{\text{ref}}$ serves as a baseline to quantify the improvement in alignment achieved by the DPO teacher $\pi\_{\text{dpo}}$. The advantage function: $A\_{\text{dpo}}(s\_t, a\_t)$ measures how much more preferable an action is compared to the reference policy. This relative measure is crucial because it highlights the specific adjustments made during preference alignment.
>
> **If we remove $\pi\_{\text{ref}}$ and use $\pi\_{\text{dpo}}$ alone**, the training objective becomes minimizing the reverse cross-entropy between the student model and the DPO teacher. Therefore, student models would attempt to mimic the teacher's policy without understanding which aspects are aligned with human preferences. In our experiments, we observed that removing $\pi\_{\text{ref}}$ from the advantage function leads to diminished performance, as shown in an ablation study (Table 2: ADPA w/o Ref teacher).
>
> **If we remove $\pi\_{\text{dpo}}$ and use $-\pi\_{\text{ref}}$ alone**, the student model's objective becomes maximizing the reverse cross-entropy with the reference teacher. This encourages the student model to produce outputs that diverge as much as possible from the reference teacher's outputs, actively pushing it away from the reference teacher without positive guidance on where to move towards. Without the positive guidance provided by $\pi\_{\text{dpo}}$, the supervised fine-tuning loss term $\mathcal{L}\_{\text{SFT}}$ alone is insufficient to regularize the training process. Consequently, this approach is prone to training instability or even collapse, as the student model lacks direction on which actions are preferable and may deviate in unintended ways, leading to degraded performance.  In practice, we found that **student models trained in this setting often produce repetitive or meaningless content**. Therefore, we did not show their performance on the test set.

---

> > ### Comment · Reviewer_3nf4 · 2024-11-23
> >
> > Thank you for the response. I think the current rating is appropriate.

---

### Official Review · Reviewer_eV8C · 2024-11-03

**Soundness:** 3
**Presentation:** 3
**Contribution:** 3
**Rating:** 8
**Confidence:** 3

**Summary:**

This works combines the concepts of RLHF and knoledge distillation to propose the advantage-guided distillation method for preference alignment. This realizes the impressive the performance improvement of the small models in the preference alignment stage.

**Strengths:**

1. The idea of improving the performance of small model in the preference alignment stage is interesting.
2. Advantage-guided distillation from the preference-aligned teacher model to the student model is novel for knowledge distillation.
3. The experiments are detailed and the presentation is good.

**Weaknesses:**

This is a good work. The proposed method is simple yet effective. I do not have additional concerns for it.

**Questions:**

1. Why experiment on  H2O-Danube3-500M and H2O-Danube2-1.8B-Base, given the availability of more popular models today, such as LLaMA-3.2-1B, Qwen2.5-0.5B . If the experiment on more popular models can be provided, the result will be more solid.

---

> ### Author Response · Authors · 2024-11-21
> **Response to Reviewer eV8C**
>
> We appreciate your encouraging remarks and are glad that you find our work interesting and effective.
>
> **Question: Experiments on more popular models.**
>
> **Response:**
> Thank you for this valuable suggestion. In the revised paper, we have added experiments using **LLaMA-3.2-1B** as the student model and **LLaMA-3.1-8B** as the teacher model. We chose LLaMA-3.2-1B over other models like Qwen-0.5B due to its wider usage in the research community.
>
> Below, we present the evaluation results for LLaMA-3.2-1B trained on DPO-MIX-7K and HelpSteer2. **We can observe that our proposed ADPA+ surpasses all baseline methods, indicating that ADPA+ is a robust and generalizable approach for the preference alignment of small language models.**
>
> | **Student**       | **Dataset**   | **Method**  | **MT-Bench** | **AlpacaEval** | **Dataset**   | **Method**  | **MT-Bench** | **AlpacaEval** |
> |--------------------|---------------|-------------|--------------|---------------|-------------|--------------|-|-|
> | LLaMA-3.2-1B      | DPO-MIX-7K    | SFT         | 3.34      |35.7   | HelpSteer2    | SFT         | 3.13         | 38.7|
> | LLaMA-3.2-1B      | DPO-MIX-7K    | DPO         | 3.40      |33.2   | HelpSteer2    | DPO         | 3.38         | 39.3 |
> | LLaMA-3.2-1B      | DPO-MIX-7K    | VanillaKD   | 3.40    | 34.1     | HelpSteer2    | VanillaKD   | 3.58         | 40.2 |
> | LLaMA-3.2-1B      | DPO-MIX-7K    | SeqKD       | 3.74       |29.7  | HelpSteer2    | SeqKD       | 3.44         | 44.4 |
> | LLaMA-3.2-1B      | DPO-MIX-7K    | ATKD        | 3.62      |32.4   | HelpSteer2    | ATKD        | 3.59         | 42.2|
> | LLaMA-3.2-1B      | DPO-MIX-7K    | PLAD        | 3.42     |29.3    | HelpSteer2    | PLAD        | 3.36         | 37.8|
> | LLaMA-3.2-1B      | DPO-MIX-7K    | DDPO        | 3.21    |28.7     | HelpSteer2    | DDPO        | 3.34         | 37.3 |
> | LLaMA-3.2-1B      | DPO-MIX-7K    | DPKD        | 3.29     |28.9    | HelpSteer2    | DPKD        | 3.10         | 36.5 |
> | LLaMA-3.2-1B      | DPO-MIX-7K    | DCKD        | 3.50     |37.5    | HelpSteer2    | DCKD        | 3.44         | 40.7 |
> | LLaMA-3.2-1B      | DPO-MIX-7K    | ADPA        | *3.88*    | *50.0*     | HelpSteer2    | ADPA        | *3.62*         | *50.0* |
> | LLaMA-3.2-1B      | DPO-MIX-7K    | ADPA+       | **4.02**  | **53.8**   | HelpSteer2    | ADPA+       | **3.99**     | **60.9**|

---

> > ### Comment · Reviewer_eV8C · 2024-11-24
> >
> > Thanks for your response. I will keep my rating for the paper.

---

### Official Review · Reviewer_346S · 2024-11-03

**Soundness:** 3
**Presentation:** 3
**Contribution:** 3
**Rating:** 8
**Confidence:** 4

**Summary:**

This paper studies the alignment of small language models. It points out that directly applying alignment techniques to small language models may not work well due to the limited capacity of these models. To this end, the authors present 1) a straightforward approach Dual-Constrained KD by integrating both positive and negative signals, and 2) an enhanced approach Advantage-Guided Distillation for Preference Alignment that involves a distribution-0level reward signal given by an advantage function. Extensive experiments with three teacher-student model settings demonstrate the effectiveness of the proposed method.

**Strengths:**

1. This paper investigates the challenge of aligning small language models, the proposed methods are well-motivated and principal.

2. The ADPA method considers the distribution-level reward signal, which is an advancement compared to the previous KD methods.

3. The KD baselines are comprehensively compared, and the studied teacher-student settings are representative.

4. This paper is well organized and the presentation is decent.

**Weaknesses:**

1. Limited baseline of alignment methods. In Table 1, while the KD baselines are comprehensively compared, the alignment baseline only includes DPO.

2. Computation overhead of the KD methods. It is not clear whether the proposed methods (and other baseline KD methods) consume more training or inference resources compared to directly applying alignment techniques to the student models.

**Questions:**

1. I wonder if directly applying some advanced alignment methods to the smaller language models can present a competitive performance.

2. Could you elaborate more on the computational cost of the proposed methods v.s. directly applying alignment techniques?

---

> ### Author Response · Authors · 2024-11-21
> **Response to Reviewer 346S**
>
> Thank you for your positive assessment and constructive suggestions regarding our work.
>
> **W1: Limited baselines of alignment methods; only DPO is included as an alignment baseline.**
>
> **Response:**
> We sincerely appreciate your insightful feedback regarding the selection of alignment baselines in our study. To clarify, our original submission included additional baselines beyond DPO. Specifically, we evaluated DPKD (utilizing the teacher model as the reference in DPO), DDPO (aligning implicit rewards derived from the student model with those from the teacher model), and PLAD (leveraging teacher-student generated pseudo-preference pairs). These methods investigate diverse dimensions of alignment through knowledge distillation or model guidance, and are closely related to the preference alignment objective of our work.
>
> We fully agree that incorporating more recent preference alignment techniques can further strengthen the robustness of our comparative analysis. In response to your suggestion, we have extended our experiments to include the latest preference alignment methods, such as **Simple Preference Optimization (SimPO) [1]** and **Weighted Preference Optimization (WPO) [2]**. Our updated empirical results indicate that while these advanced preference alignment approaches yield moderate improvements in alignment metrics over the DPO baseline, our proposed ADPA framework consistently demonstrates more substantial performance gains.
>
> | **Policy Model** | **Dataset**     | **Method** | **MT-Bench** | **AlpacaEval** |
> |-|-|-|-|-|
> | Danube2-1.8B        | DPO-MIX-7K      | DPO        | 3.87         |  30.6 |
> | Danube2-1.8B        | DPO-MIX-7K      | SimPO      | 3.97         | 16.8 |
> | Danube2-1.8B        | DPO-MIX-7K      | WPO        | 4.03         | 31.8|
> | Danube2-1.8B        | DPO-MIX-7K      | ADPA       | *4.33*         | *50.0* |
> | Danube2-1.8B        | DPO-MIX-7K      | ADPA+      | **4.74**         | **61.3** |
> | Danube2-1.8B        | HelpSteer2      | DPO        | 3.87         | 40.7 |
> | Danube2-1.8B        | HelpSteer2      | SimPO      | 3.95         | *58.4* |
> | Danube2-1.8B        | HelpSteer2      | WPO        | *4.23*         | 54.5 |
> | Danube2-1.8B        | HelpSteer2      | ADPA       | 4.02         | 50.0 |
> | Danube2-1.8B        | HelpSteer2      | ADPA+      | **4.33**         | **62.4** |
>
> **References:**
> [1] SimPO: Simple preference optimization with a reference-free reward. 2024. In NeurIPS.
> [2] WPO: Enhancing RLHF with Weighted Preference Optimization. 2024. In EMNLP.
>
> ---
>
> **W2: Computational overhead of the KD methods compared to directly applying alignment techniques.**
>
> **Response:**
> We appreciate the reviewer’s insightful comment regarding the computational overhead of KD-based alignment methods compared to directly applying alignment techniques.
>
> To address this, we provide the following clarifications. First, while KD methods involve the use of additional teacher models during training, the associated computational cost can be mitigated by precomputing the required teacher outputs offline before training begins. Second, once the student model is trained, the inference stage operates without incurring any additional computational overhead. **Third, importantly, the primary challenge in aligning small language models lies not in the computational cost during training, but in overcoming the inherent limitations in their capacity and capabilities.** Therefore, KD has been widely used as a crucial technique to address these challenges. Notable examples include its application in models such as LLaMA3.2 [3] and Gemma2 [4].
>
> **References:**
> [3] LLaMA 3.2: Revolutionizing edge AI and vision with open, customizable models. Meta AI.
> [4] Gemma 2: Improving Open Language Models at a Practical Size. Gemma Team, Google DeepMind.

---

> > ### Comment · Reviewer_346S · 2024-11-27
> >
> > Thank you for your response. I believe the current rating is appropriate.

---

> ### Author Response · Authors · 2024-11-25
> **Looking Forward to the Opportunity for Further Discussion**
>
> Dear Reviewer 346S,
>
> We sincerely appreciate the time and effort you have devoted to reviewing our work, especially considering your busy schedule. As the discussion phase between authors and reviewers draws to a close, we kindly request your attention to our responses. In light of your feedback, we have incorporated additional alignment baselines, including recent methods like SimPO and WPO, and found that our ADPA framework consistently outperforms them. Regarding your concern about computational overhead, we have clarified that our method involves minimal additional computations during training, which can be mitigated through offline precomputations, and there is no added cost during inference.
>
> We hope that we have effectively addressed your concerns and would welcome any additional questions or points you might like to discuss. We highly value your feedback and look forward to further discussion.
>
> Thank you for your thoughtful consideration.
>
> Best regards,
>
> The Authors

---

### Official Review · Reviewer_PkN8 · 2024-11-04

**Soundness:** 3
**Presentation:** 3
**Contribution:** 3
**Rating:** 8
**Confidence:** 3

**Summary:**

The paper explores how to enhance the effectiveness of small language models  to make their generated outputs more aligned with human preferences through preference alignment techniques. To address the issue that technologies like RLHF do not align well with human preferences on small language models, the paper proposes two methods: Dual-Constrained Knowledge Distillation (DCKD) and Advantage-Guided Distillation for Preference Alignment (ADPA). The experimental results show that both methods can significantly improve the alignment of small language models and narrow the performance gap with large models.

**Strengths:**

- The two methods proposed in the paper combine RLHF with distillation, providing insights into the preference alignment of small language models and solving problems that previous methods had not addressed.
- The experimental section is well-designed, with numerous comparative and ablation experiments to verify the performance improvements of the methods.
- The paper is well-written, with detailed descriptions of the two methods, including algorithm steps and formulas, which are easy to understand.

**Weaknesses:**

- ADPA provides more nuanced guidance signals, but the additional computations introduced by fine-grained signals may increase the computational overhead, and the paper seems to lack specific quantitative metrics on this point.
- The experimental section utilizes a rather singular evaluation dataset, without providing results on a broader range of models, domains, and data. Additionally, there is a lack of experiments assessing the impact of teachers of varying proficiency levels on the results.

**Questions:**

- In Formula 11, $L_s$ seems to be undefined. Does it refer to $L_{SFT}$?

---

> ### Author Response · Authors · 2024-11-21
> **Response to Reviewer PkN8 (part 1)**
>
> We appreciate your positive feedback and thoughtful review of our work.
>
> **W1: Computational overhead introduced by fine-grained signals.**
>
> **Response:** We appreciate the reviewer's concern regarding the computational overhead introduced by the fine-grained reward signals in our ADPA framework. We would like to clarify that the training process of ADPA does not significantly increase computational overhead, especially when compared to traditional knowledge distillation (KD).
>
> Specifically, the computation involving the fine-grained reward signals in ADPA requires calculating $E\_{a\_t \sim \pi\_{\text{stu}}(\cdot|s\_t)} \left( \log \pi\_{\text{dpo}}(a\_t|s\_t) - \log \pi\_{\text{ref}}(a\_t|s\_t) \right),$ which is somewhat similar to the KLD objective commonly employed in traditional KD. Both objectives require computing an expectation over the difference between two log probability distributions. Although ADPA introduces an additional teacher, the computational cost associated with this is minimized through precomputation of these values offline prior to training.
>
> During our experiments training a h2o-danube2-1.8B student model on the DPO-MIX-7K dataset using 8 RTX 3090 GPUs with a per-device batch size of 8 (global batch size of 128) and DeepSpeed ZeRO-2, APKD resulted in only a slight increase in both memory usage and training time, compared to Vanilla KD. Specifically, **memory usage rose by less than 3%** (from 21,957 MB to 22,541 MB), and **training time increased by less than 4%** (from 78 minutes to 81 minutes). These minor overheads are well within acceptable limits for practical applications.
>
> Thus, the overall impact on computational resources remains minimal and well within acceptable bounds.
>
> ---
>
> **W2: Utilizing a rather singular evaluation dataset; lack of results on a broader range of models, domains, and data.**
>
> **Response:** Thank you for this valuable feedback. To address this concern, we have expanded our experimental scope by incorporating additional models and evaluation datasets, as outlined below:
>
> 1. **Additional Models:** We have included experiments with an additional student model, LLaMA-3.2-1B, trained under the guidance of the teacher model LLaMA-3.1-8B on the DPO-MIX-7K dataset. This addition aims to demonstrate the adaptability of our proposed method to popular models widely utilized within the research community.
>
> 2. **Broader Evaluation Datasets:** To ensure a more comprehensive assessment, we evaluated our trained models on the Open LLM Leaderboard, which benchmarks open-source language models across six diverse domains, including scientific reasoning, commonsense reasoning, multi-task understanding, and factual accuracy. This broader evaluation provides a robust analysis of our approach's effectiveness across various tasks.
>
> We present the evaluation results for LLaMA-3.2-1B on MT-Bench, AlpacaEval (ADPA as reference) and the Open LLM Leaderboard below. Our proposed methods, ADPA and ADPA+, consistently outperform all baseline approaches across most benchmarks, demonstrating exceptional performance on ARC-Challenge, HellaSwag, and TruthfulQA. This underscores their effectiveness in tackling diverse tasks. These results will be included in the revised manuscript for clarity and completeness.
>
> | **Student** | **Method**  | **MT-Bench** | **AlpacaEval** |**ARCC** | **HellaSwag** | **TruthfulQA** | **MMLU** | **Winogrande** | **GSM8k** | **Open LLM Leaderboard Average** |
> |-|-|-|-|-|-|-|-|-|-|-|
> | LLaMA-3.2-1B  | SFT  | 3.34| 35.7 | 41.04| 68.38 | 40.91 | 34.38    | 60.46 | 6.82 | 42.00    |
> | LLaMA-3.2-1B  | DPO  | 3.4| 33.2| 40.27| 68.51 | 45.17 | 34.38    | 61.25 | 6.60 | 42.70    |
> | LLaMA-3.2-1B  | VanillaKD   | 3.4| 34.1| 41.04| 67.24 | 42.91 | 34.04    | **62.90** | *7.05* | 42.53    |
> | LLaMA-3.2-1B  | SeqKD  | 3.74| 29.7 | 40.02| 66.49 | 44.11 | **35.28**    | 61.25 | 5.84 | 42.17    |
> | LLaMA-3.2-1B  | ATKD | 3.62| 32.4 | 40.27| 67.33 | 44.66 | 34.38    | 60.77 | 6.29 | 42.28    |
> | LLaMA-3.2-1B  | PLAD | 3.42| 29.3 | 39.93| 68.22 | 44.55 | 34.42    | 61.56 | 5.16 | 42.31    |
> | LLaMA-3.2-1B  | DDPO | 3.21| 28.7 | 39.51| 66.84 | 44.55 | 33.91    | 61.72 | 5.61 | 42.02    |
> | LLaMA-3.2-1B  | DPKD | 3.29| 28.9 | 38.74| 66.18 | 45.18 | 34.57    | 61.32 | 5.23 | 41.87    |
> | LLaMA-3.2-1B  | DCKD | 3.5 | 37.5 | 41.21| 67.64 | 43.38 | 34.57    | *61.80* | **7.51** | 42.69    |
> | LLaMA-3.2-1B  | ADPA | *3.88* | *50.0* | *41.81*| *68.66* | **47.98** | *34.74*| *61.80* | 5.31 | **43.38**    |
> | LLaMA-3.2-1B  | ADPA+| **4.02** | **53.8**| **42.15**| **70.49** | *45.89* | 34.60    | 59.91 | 5.16 | *43.03*    |
>
> ---

---

> ### Author Response · Authors · 2024-11-21
> **Response to Reviewer PkN8 (part 2)**
>
> **W2 (the latter part): Absence of experiments assessing the impact of teachers of varying proficiency levels.**
>
> **Response:** We appreciate the reviewer's insightful comment regarding the impact of teachers with varying proficiency levels on small language (student) models. To address this, we conducted additional experiments to directly assess this factor. Specifically, we compared the effects of two teacher models with distinct proficiency levels: H2o-danube2-1.8B, representing a lower-proficiency teacher, and Mistral-7B, a higher-proficiency teacher previously used in our experiments. In both cases, the student model was H2o-danube3-0.5B, and all experiments were performed on the DPO-MIX-7K dataset.
>
> The empirical results are summarized in the following table. We observe that the teacher model's proficiency influences knowledge transfer results. Notably, higher-proficiency teacher models, such as Mistral-7B, consistently enable superior performance in student models through the distillation process. These results validate the critical role of teacher proficiency in knowledge transfer and will be included in the revised manuscript for clarity and completeness.
>
> | **Student**   | **Teacher Model**   | **KD Method** | **MT-Bench** | **AlpacaEval**|
> |-|-|-|-|-|
> | H2O-Danube3-0.5B| H2O-Danube2-1.8B   | VanillaKD| 2.55  | 37.8 |
> | H2O-Danube3-0.5B| H2O-Danube2-1.8B   | SeqKD  | 2.61  | 36.6 |
> | H2O-Danube3-0.5B| H2O-Danube2-1.8B   | DCKD | 2.67  | 39.8 |
> | H2O-Danube3-0.5B| H2O-Danube2-1.8B   | ADPA | *2.71*  | *50.0*|
> | H2O-Danube3-0.5B| H2O-Danube2-1.8B   | ADPA+  | **2.75**  | **54.2** |
> | H2O-Danube3-0.5B| Mistral-7B  | VanillaKD| 2.60  | 39.5 |
> | H2O-Danube3-0.5B| Mistral-7B  | SeqKD  | 2.53  | 39.3 |
> | H2O-Danube3-0.5B| Mistral-7B  | DCKD | *2.77*  | 39.2 |
> | H2O-Danube3-0.5B| Mistral-7B  | ADPA | 2.67  | **50.0** |
> | H2O-Danube3-0.5B| Mistral-7B  | ADPA+  | **2.78**  | *48.3* |
>
> ---
>
> **Q1: In Formula 11, $L\_s $ seems to be undefined. Does it refer to $L\_{\text{SFT}}$?**
>
> **Response:** Thank you for pointing this out. You are correct that in Equation (11), $L\_s $ refers to the supervised fine-tuning loss $\mathcal{L}\_{\text{SFT}}$. We apologize for the oversight in the notation. In the revised manuscript, we have thoroughly reviewed and corrected the notation to ensure all terms are clearly defined and consistent throughout. Specifically, we have replaced $L\_s $ with $\mathcal{L}\_{\text{SFT}} $ in Equation (11) and throughout the manuscript where applicable.

---

> ### Author Response · Authors · 2024-11-25
> **Looking Forward to the Opportunity for Further Discussion**
>
> Dear Reviewer PkN8,
>
> We sincerely appreciate the time and effort you have devoted to reviewing our work, especially given your undoubtedly busy schedule. As the discussion phase between authors and reviewers draws to a close, we kindly invite you to review our responses to your comments.
>
> Our primary aim has been to address your feedback comprehensively and to engage further on any remaining questions or suggestions you may have. Your insights have been invaluable, and we are eager to continue this constructive dialogue.
>
> To address your comments and strengthen our findings, we have undertaken the following actions:
> 1. Demonstrated that the additional cost is minimal.
> 2. Expanded our experiments to include additional models, such as LLaMA-3.2-1B, and evaluated them on the OpenLLM Leaderboard.
> 3. Conducted experiments with teacher models of varying proficiency levels, which reaffirmed the critical role of teacher quality in effective knowledge transfer.
>
> We sincerely thank you once again for your thoughtful feedback and careful consideration. We look forward to any further insights you may have.
>
> Best regards,
> The Authors

---

> > ### Comment · Reviewer_PkN8 · 2024-11-25
> >
> > Thanks for your response. All the questions I raised have been well resolved, and I have updated the rating accordingly.

---

### Meta-Review · Area_Chair_dMTM · 2024-12-21

**Metareview:**

This paper studies alignment for small language models. The motivation is that we’d prefer to use small language models for efficiency reasons, but standard alignment approaches directly applied to these struggle. Instead, the authors propose a distillation-style approach where a (larger) aligned teacher is used to guide the small model.

Both the motivation for the problem and the approach are nice. The results are quite strong.

There are no strong weaknesses here; the reviewers brought up questions around the clarity of the writing, but these were addressed by the authors during rebuttal. Another potential question is whether the results hold up for other small models (or more generally, other combinations of teacher and student).

Overall, this is a solid work and worth accepting.

**Additional Comments On Reviewer Discussion:**

Reviewers generally asked about aspects of the writing, baselines, and experimental details. The authors clarified everything successfully.

---

### Decision · Program_Chairs · 2025-01-22

Accept (Spotlight)